# Membrane curvature governs the distribution of Piezo1 in live cells

**Shilong Yang[1], Xinwen Miao[2,9], Steven Arnold[3,9], Boxuan Li[1,9], Alan T. Ly ⓘ [4,5], Huan Wang[1], Matthew Wang[6], Xiangfu Guo[2], Medha M. Pathak ⓘ [4,5], Wenting Zhao ⓘ [2], Charles D. Cox ⓘ [7] & Zheng Shi ⓘ [1,8] ✉**

Piezo1 is a bona fide mechanosensitive ion channel ubiquitously expressed in mammalian cells. The distribution of Piezo1 within a cell is essential for various biological processes including cytokinesis, cell migration, and wound healing. However, the underlying principles that guide the subcellular distribution of Piezo1 remain largely unexplored. Here, we demonstrate that membrane curvature serves as a key regulator of the spatial distribution of Piezo1 in the plasma membrane of living cells. Piezo1 depletes from highly curved membrane protrusions such as filopodia and enriches to nanoscale membrane invaginations. Quantification of the curvature-dependent sorting of Piezo1 directly reveals the in situ nano-geometry of the Piezo1-membrane complex. Piezo1 density on filopodia increases upon activation, independent of calcium, suggesting flattening of the channel upon opening. Consequently, the expression of Piezo1 inhibits filopodia formation, an effect that diminishes with channel activation.

Piezo1 is a widely expressed mechanosensitive ion channel in the plasma membrane of eukaryotic cells[1,2], crucial for a broad range of mechanotransduction processes[3–5]. Each Piezo1 monomer contains 38 transmembrane helices and cryogenic electron microscopy (CryoEM) shows a trimeric propeller-like assembly which is thought to curve into the cytosol in its resting state[6–9]. Recently, a flattened configuration of Piezo1 was identified when reconstituted into small liposomes, potentially corresponding to the open/inactivated state of the ion channel[10] and confirming previous work using high speed atomic force microscopy (HS-AFM)[11]. The large size and curved architecture of Piezo1 trimers make them directly sensitive to tension in the plasma membrane[7,12–14]. Additionally, several studies indicate a cytoskeletal role in the activation of Piezo channels, while the presence of direct Piezo1-cytoskeleton interaction is still under debate[15–22]. Notably, the cortical cytoskeleton can drastically affect the extent of membrane tension propagation, thereby indirectly controlling Piezo1 activation via the lipid bilayer[23–26].

While the structure and activation of individual Piezo1 channels have been extensively studied, the dynamics and distribution of the channel within a cell are only starting to be explored[19,21,25,27]. In addition, it is unclear whether the Piezo1 structures determined in vitro recapitulate its nanoscale conformations in living cells[7,8,10,11]. Several recent studies highlighted a preferential subcellular distribution of Piezo1: First, a polarized distribution of Piezo1 towards the rear of migrating keratinocytes plays a crucial role in controlling the speed of wound healing[28]. Additionally, Piezo1 has been found to enrich at subcellular structures that are important for mechanotransduction, such as focal adhesion sites[27,29] and T-tubules of cardiomyocytes[30,31]. Lastly, Piezo1 has been reported to preferentially localize to the intercellular bridge during cytokinesis[32], and to the biconcave 'dimples' in red blood cells[19,21]. While a general mechanistic explanation has yet to be established, these intriguing subcellular patterns of Piezo1 raise the question of whether this ion channel can be sorted by fundamental physical factors on the cell surface.

[1]Department of Chemistry and Chemical Biology, Rutgers University, Piscataway, NJ 08854, USA. [2]School of Chemistry, Chemical Engineering, and Biotechnology, Nanyang Technological University, 637457 Singapore, Singapore. [3]Department of Cell Biology and Neuroscience, Rutgers University, Piscataway, NJ 08854, USA. [4]Department of Physiology & Biophysics, UC Irvine, Irvine, CA 92697, USA. [5]Sue and Bill Gross Stem Cell Research Center, UC Irvine, Irvine, CA 92697, USA. [6]Department of Physics, Rutgers University, Piscataway, NJ 08854, USA. [7]Molecular Cardiology and Biophysics Division, Victor Chang Cardiac Research Institute, Sydney, NSW, Australia. [8]Cancer Pharmacology Research Program, Cancer Institute of New Jersey, Rutgers University, New Brunswick, NJ 08901, USA. [9]These authors contributed equally: Xinwen Miao, Steven Arnold, Boxuan Li. ✉e-mail: zheng.shi@rutgers.edu

Several hints in the literature indicate that membrane curvature may play a role in regulating the cellular distribution of Piezo1. First, the structure of purified Piezo1 trimers shows that they form nanoscale invaginations in a simple liposomal system[7,11]. When locally stretching membrane tethers to activate Piezo1, calcium ($Ca^{2+}$) influx initiates around the tether-cell attachment point where membrane tension is high[23]. However, $Ca^{2+}$ initiation sites are noticeably missing from the highly tensed membrane tether itself[23]. More surprisingly, a recent study showed that force dependent $Ca^{2+}$ signals in filopodia are independent of Piezo channels[33]. Filopodia are highly curved membrane protrusions that geometrically resemble the artificially pulled membrane tethers (tube of radius ~50 nm). Thus, a simple explanation would be that Piezo1 are actually absent from these highly curved membrane protrusions.

Here, we combine high-throughput filopodia[34] and nanobar[35,36] sorting assays with quantitative single membrane tether pulling experiments[23] to show that membrane curvature is a fundamental regulator of Piezo1's distribution within the plasma membrane. The curvature mismatch between Piezo1 and membrane protrusions prevents the channel from entering structures such as filopodia and thin membrane tethers. In contrast, Piezo1 significantly enriches to curved cell membrane invaginations induced by engineered nanobars[35,36]. Quantifying the curvature preference of Piezo1 on a wide range of tether radii reveals the nano-geometry of Piezo1-membrane complex in living cells. Furthermore, a chemical activator of Piezo1, Yoda1, which has been hypothesized to serve as a molecular wedge to bias the protein towards a less-curved state[37], allows Piezo1 to enter filopodia in a $Ca^{2+}$-independent manner. The curvature-preference and Yoda1-response of Piezo1 sorting in cells are consistent with recently determined structural features of purified Piezo1 trimers[10]. The coupling between curvature dependent sorting and activation of Piezo1 in living cells likely represents a fundamental cornerstone of Piezo1 channel biology, enabling the regulation of filopodia formation and retaining Piezo1 in the cell body during cell migration.

## Results

### Piezo1 is depleted from filopodia

To study the distribution of Piezo1 in the plasma membrane, we first co-expressed human Piezo1 (hPiezo1-eGFP[12]) and glycosylphosphatidylinositol (GPI) anchored mCherry in HeLa cells. Piezo1 traffics well to the plasma membrane, as indicated by an eGFP fluorescence profile across the cell body that closely resembles the co-expressed plasma membrane marker (Figs. 1a and S1). However, the Piezo1 signal was noticeably missing on filopodia that protrudes around the edge of the cell, in drastic contrast with membrane markers such as GPI and CaaX, and with other transmembrane proteins such as the dopamine receptor D2 (D2R) and the mechanosensitive potassium channel TREK1 (Figs. 1a−c and S2). To systematically quantify protein densities on filopodia, we defined a unitless filopodia sorting value ($S_{filo}$) using the fluorescence ratio between the molecule of interest (MOI) and the membrane reference (Fig. S3, Methods). This fluorescence ratio measured along a filopodium is normalized to the same ratio on a flat region of the cell body to account for cell-cell variabilities. Additionally, the well-defined membrane geometry on a flat region of the cell allows us to directly quantify the diffraction-limited radii of filopodia from the fluorescence of membrane markers[23].

We found the $S_{filo}$ of Piezo1 is close to 0, significantly smaller than the sorting of other MOIs ($S_{filo}$ ~1; Fig. 1d). The lack of Piezo1 on filopodia is independent of imaging temperature, the choice of fluorescent protein (FP) tag, FP fusion position, or Piezo1 species (Fig. 1d). In addition to HeLa cells, Piezo1 is depleted from the filopodia of HEK293T (Fig. S4) and from the filopodia of mouse embryonic fibroblasts (MEFs) where tdTomato-labelled Piezo1 is expressed at endogenous levels (Fig. S5)[25,28]. Referencing to the endogenous Piezo1-tdTomato fluorescence in MEFs, the amount of overexpressed Piezo1 in HeLa cells is

estimated to be $2.5 \pm 1.5$ (mean ± SD, $n = 12$) fold (Fig. S5c). Notably, D2R and TREK1 are significantly enriched on filopodia ($S_{filo} > 1$). The filopodia enrichment of D2R agrees with established membrane curvature preference of GPCRs[34], whereas the enrichment of TREK1 potentially reflects the protein's role in filopodia formation[38]. Cells that overexpress Piezo1 had the same filopodia radii as cells expressing membrane markers (Fig. 1e). In contrast, cells overexpressing D2R or TREK1 showed significantly reduced filopodia radii, consistent with the membrane deformation and curvature generation ability of many curvature sensing proteins[39,40]. However, the molecular mechanisms and the causalities between the increased $S_{filo}$ and the reduced filopodia radii in D2R or TREK1 expressing cells remain to be explored.

The absence of Piezo1 on filopodia is consistent with the dispensability of Piezo1 for mechanically activated $Ca^{2+}$ signals in filopodia[33]. Additionally, the quantified curvature preference of Piezo1 ($S_{filo} = 0.027 \pm 0.003$, mean ± SEM) is in accord with a recent CryoEM observation that only ~3% of purified Piezo1 trimers were oriented 'outside-out' (extracellular domains of Piezo1 facing the outside of liposomes), as a Piezo1 trimer on filopodia would be, when reconstituted into highly curved liposomes[10].

### Depletion of Piezo1 is not specific to filopodia and is independent of cytoskeleton

Cellular protrusions such as filopodia typically contain a complex network of actin-rich structures[41]. Is the observed depletion of Piezo1 in Fig. 1 specific to filopodia? To answer this question, we focused our further investigation on membrane tethers that geometrically mimic filopodia but lack specific actin-based structures when freshly pulled[23] (Fig. 2a).

Similar to the observation on filopodia, tethers pulled from HeLa cells or MEFs only contained signal for the membrane marker (Figs. 2b and S5b). There is no significant difference between Piezo1's sorting on tethers ($S_{teth}$) and on filopodia ($S_{filo}$) (Fig. 2e). The geometrical similarity between tethers and filopodia (both are highly curved membrane protrusions) points to a possible role of membrane curvature in mediating the sorting of Piezo1. However, $S_{filo}$ did not show any apparent dependence on filopodium radius, unlike that of D2R or TREK1 (Fig. S6). Additionally, $S_{teth}$ is independent of the relaxation of pulled tethers (Fig. S7). Notably, filopodia only present a small range of membrane radii (25-50 nm). Radii of short and fully relaxed tethers are similar to those of filopodia, while stretched tethers are typically thinner[23] (Fig. 2f). Therefore, we hypothesize that the sorting of Piezo1 is most sensitive to membrane curvatures corresponding to >50 nm radii protrusions. Alternatively, it is plausible that Piezo1 is strongly attached to the cortical cytoskeleton, preventing the channel from moving onto membrane protrusions.

To test the two hypotheses, we generated stable membrane blebs via pharmacological depolymerization of the actin cytoskeleton[23]. Similar to previous reports[12], bleb membranes clearly contain Piezo1 signal, but not significantly enriched relative to the cell body (Figs. 2c, d and S8). More importantly, negligible Piezo1 fluorescence was observed on membrane tethers pulled from tense blebs (Fig. 2c). Plasma membrane blebs do not contain cytoskeleton, therefore results such as Fig. 2c directly argue against a main cytoskeletal role in the depletion of Piezo1 from tethers. The radius of a tether pulled from the bleb is determined by the bleb's membrane tension[23], which is in turn governed by the intracellular pressure[42]. We found that membrane blebs triggered by actin depolymerization exhibited a wide range of apparent 'floppiness', likely a result of stochastic pressure release during bleb formation. On floppy (i.e., low membrane tension) blebs, pulled tethers showed much wider apparent radii (Fig. 2d; Eq. 7). Importantly, Piezo1 fluorescence can be clearly observed on these wide tethers, leading to a highly scattered $S_{teth}$ of Piezo1 on tethers pulled from blebs (Fig. 2e). Tethers are typically imaged >1 min after pulling, whereas membrane tension equilibrates within 1 s across cellular scale

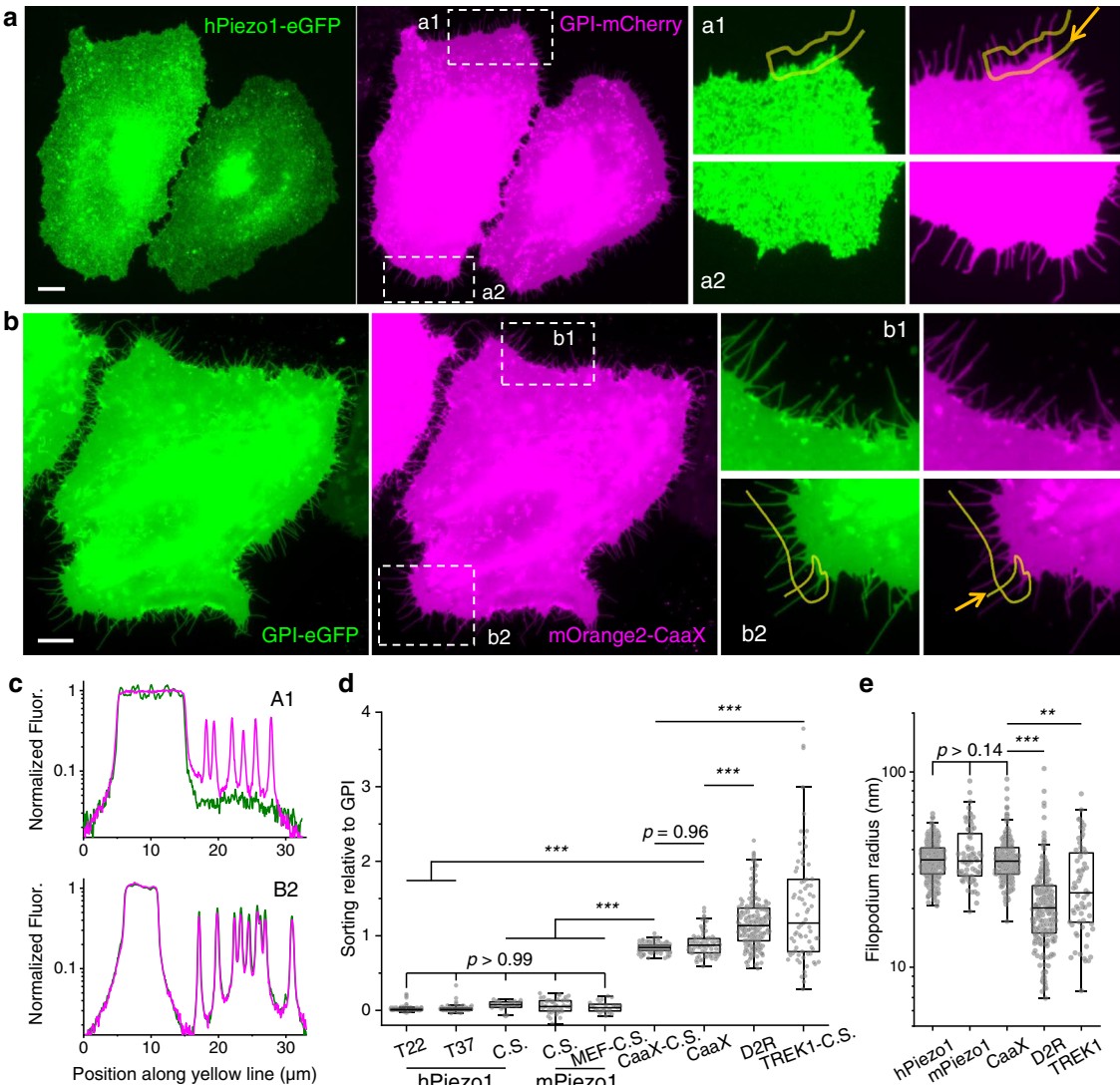

**Fig. 1 | Piezo1 is depleted from filopodia. a** Fluorescence images of HeLa cells co-expressing hPiezo1-eGFP (left) and GPI-mCherry (middle). The two boxed regions (a1, a2) are zoomed-in on the right. Images are contrast-adjusted to show the dim filopodia, see Fig. S1a for images under regular contrast. **b** Fluorescence images of HeLa cells co-expressing GPI-eGFP (left) and mOrange2-CaaX (middle). The two boxed regions (b1, b2) are zoomed-in on the right. All scale bars are 10 μm. **c** Fluorescence intensity profiles along the marked yellow lines in a1 (up) and b2 (down). Green: hPiezo1 (up) and GPI (down). Magenta: GPI (up) and CaaX (down). Each fluorescence trace was normalized to the mean intensity on the corresponding flat cell body. **d** Filopodia sorting of eGFP fused hPiezo1 (T22: at 22 °C, n.f. = 129, n.c. = 12; T37: at 37 °C: n.f. = 113, n.c. = 8), CaaX (n.f. = 87, n.c. = 9), and D2R (n.f. = 222, n.c. = 15) relative to GPI-mCherry. C.S.: color swap, indicating the molecule of interest was fused with red or orange fluorescent proteins (tdTomato for

MEF, mOrange2 for CaaX, mCherry for the rest) while the reference was GPI-eGFP. hPiezo1-C.S.: n.f. = 24, n.c. = 4; mPiezo1-C.S.: n.f. = 47, n.c. = 8; mPiezo1-MEF-C.S.: n.f. = 24, n.c. = 5; CaaX-C.S.: n.f. = 123, n.c. = 9; TREK1-C.S.: n.f. = 77, n.c. = 12. n.f.: number of filopodia, n.c.: number of cells. **e** Filopodia radii of cells co-expressing GPI and hPiezo1 (n.f. = 266), mPiezo1 (n.f. = 71), CaaX (n.f. = 210), D2R (n.f. = 222), or TREK1 (n.f. = 65). All radii were determined from the GPI channel. Data labelled with 'MEF' were done in mouse embryonic fibroblasts with Piezo1-tdTomato at endogenous expression level and 'TREK1' were measured in HEK293T cells, all other quantifications were done in HeLa cells. *p* values given by one-way ANOVA with *post hoc* Tukey's test. ***$p < 10^{-7}$. **$p < 10^{-3}$. In the box plots, the mid-lines represent the median, the box represents 25-75% range, and the whiskers represent the 1.5 inter-quartile range (same below). Source data are provided as a Source Data file (same below).

free membranes (e.g., bleb, tether)[23]. Therefore, the sorting of Piezo1 within individual tension-equilibrated tether-bleb systems (Fig. 2c–g) suggests that membrane curvature can directly modulate Piezo1 distribution beyond potential confounding tension effects.

When combining the Piezo1 sorting and the tether radius measurements together, a clear positive correlation can be observed (Fig. 2g). $S_{teth}$ on tense blebs is comparable to $S_{teth}$ measured on intact cells and to the fraction of 'outside-out' Piezo1 trimers when reconstituted into small liposomes[10] (Fig. 2f, g). $S_{teth}$ of Piezo1 is independent of bleb radius (Fig. 2g, inset), confirming the lack of optical artifacts induced by bleb curvature. However, the curved geometry of blebs only allowed determination of an apparent tether radius (Fig. S3). We

assumed that the average radius of filopodia and equilibrated tethers from cell body are equal to the radii of tethers from the tensest blebs, thus converting the apparent radii (in A.U.) to absolute radii (in nm) of tethers from blebs (Fig. 2f, g). The conversion was consistent with the upper bound of tether diameters set by the width of optically resolvable catenoid-shape membranes at the two ends of low-tension tethers[43] (Fig. S8c).

### Quantification of Piezo1's molecular features

The trimer of Piezo1 has been suggested to adopt an 'outside-in' dome shape in liposomal systems[7,10,11,44]. If this were to occur in cells, Piezo1 would energetically prefer certain membrane invaginations (positive

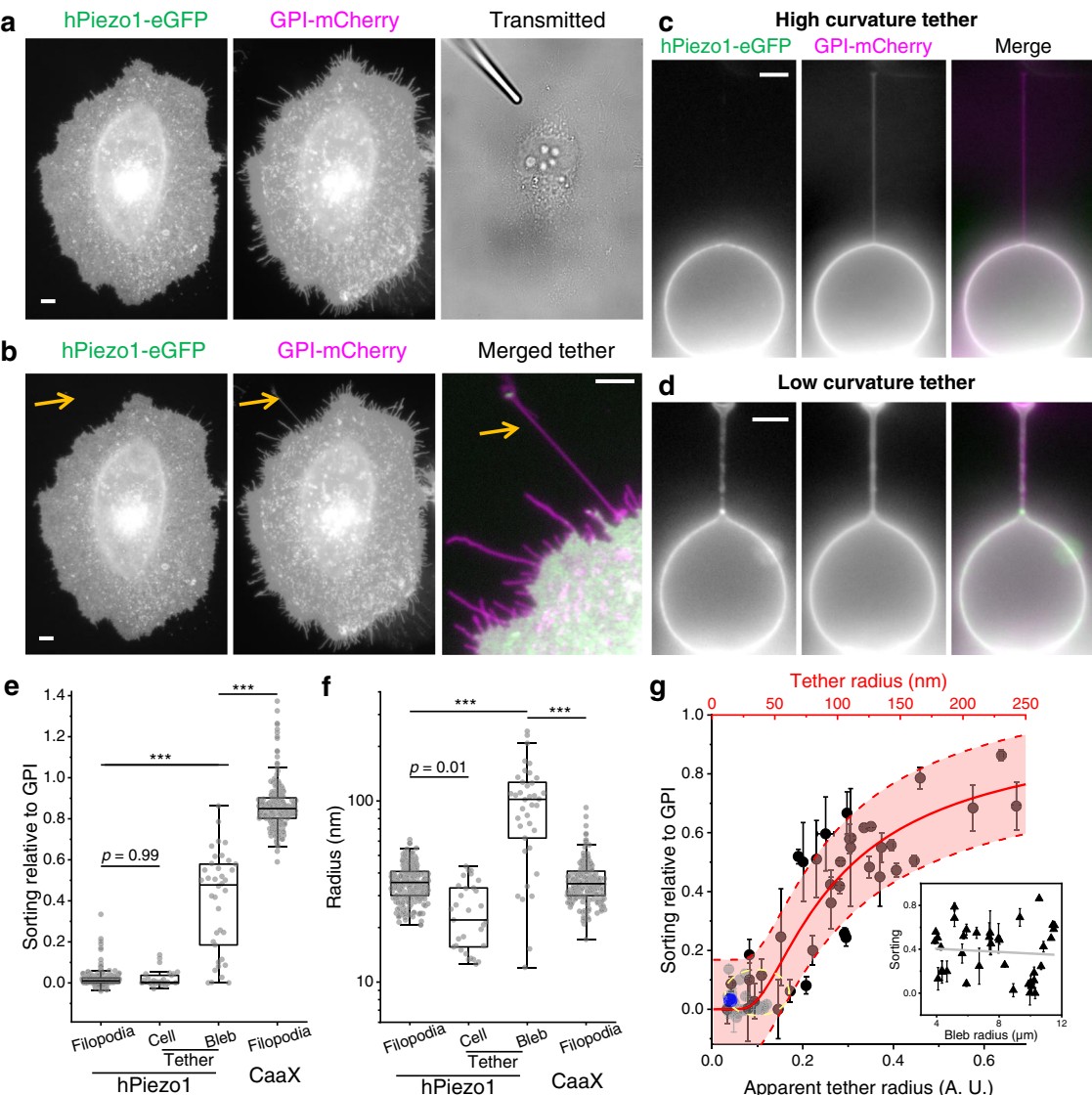

**Fig. 2 | Sorting of Piezo1 on membrane tethers. a** Fluorescence images of a HeLa cell co-expressing hPiezo1-eGFP (left) and GPI-mCherry (middle). The transmitted light image (right) shows the position of a motorized micropipette (fused-tip) in contact with the cell before tether pulling. **b** Fluorescence images of the HeLa cell in (**a**) after a 20 μm tether was pulled out (arrow). The tether region is merged and contrast-adjusted on the right. **c, d** Fluorescence images of tethers pulled from membrane blebs on HeLa cells co-expressing hPiezo1-eGFP (left) and GPI-mCherry (middle). Merged images shown on the right. Significantly less Piezo1 signals were observed on the tether from tense bleb (**c**) compared to the tether from floppy bleb (**d**). All fluorescence images are shown in log-scale to highlight the dim tether. All scale bars are 5 μm. Full images of the cell in (**c**) and (**d**) are shown in Fig. S8. **e, f** Sorting of hPiezo1 (**e**) and radii of tethers (**f**) pulled from cell membranes (Cell, $n = 31$ cells; as in **a–b**) and membrane blebs (Bleb, $n = 38$ blebs; as in **c–d**) relative to GPI. Corresponding data of hPiezo1 ($n = 266$ filopodia) and CaaX ($n = 210$ filopodia) from Fig. 1d, e are shown here for comparison. All radii were determined from the GPI channel. **g** Sorting of hPiezo1 on tethers pulled from blebs (black) plotted as a function of the apparent (lower axis) and absolute (upper axis) radii of the tethers. Sorting of hPiezo1 on tethers pulled from cells are shown in gray. The fraction of 'outside-out' Piezo1 when reconstituted into small liposomes (according to ref. 10) is shown in blue. The yellow circle shows the cluster of tense bleb data used to calculate absolute tether radii. The center and error band represent the best fit of $S_{teth}$ ($n = 69$ tethers) to Eq. (1) and the 95% confidence interval, respectively. Inset: Sorting of Piezo1 as a function of bleb radius ($n = 38$ blebs), where the line represents a linear fit with slope $= -0.007 \pm 0.015$ μm$^{-1}$. Error bars are SEM. $p$ values given by one-way ANOVA with post hoc Tukey's test. ***$p < 10^{-7}$.

curvature) and stay away from membrane protrusions (negative curvature) such as the filopodium and tether. Accordingly, we fitted our measured relation between the sorting of Piezo1 on tether ($S_{teth}$) and the tether radius ($R_t$) in Fig. 2g to a 2-parameter model based on the bending energy of membrane protrusions (Method)[34,45]:

$$S_{teth} = \exp\left[ -\widetilde{A}_P \left( \frac{1}{R_t{}^2} + \frac{2C_0}{R_t} \right) \right] \quad (1)$$

Here $\widetilde{A}_P = \frac{\widetilde{\kappa} A_P}{2k_B T}$ is the product of the unit area for each Piezo1-membrane complex ($A_P$) and its bending stiffness ($\widetilde{\kappa}$), normalized by

the Boltzmann constant ($k_B$) and the absolute temperature ($T$). $A_P$ represents the area of a potentially curved surface and should not be confused with the projected area of the protein. $C_0$, expected to be positive for Piezo1, is the spontaneous curvature of each protein-membrane complex in the (mostly flat) plasma membrane and would be lower than the apparent curvature of Piezo1 in detergents or in highly curved liposomes[10,11,44].

The two fitting parameters $\widetilde{A}_P$ and $C_0$ correspond to the contributions of Piezo1's size (larger proteins have a stronger tendency to stay away from highly curved membranes) and intrinsic curvature, respectively. The fitting gave $\widetilde{A}_P = 2400 \pm 1000$ nm$^2$ and

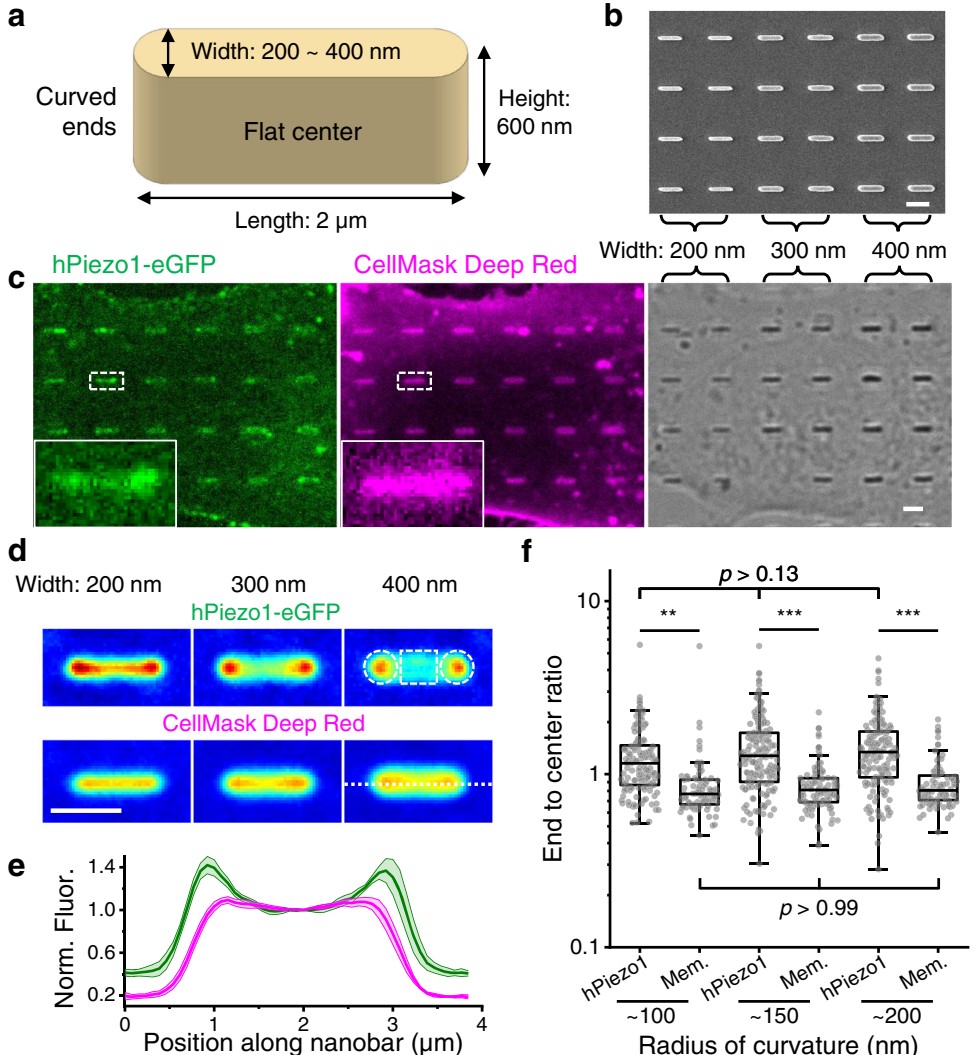

**Fig. 3 | Enrichment of Piezo1 towards the ends of nanobars. a** Illustration of a nanobar. **b** Scanning electron microscopy image of the cell culture substrate with 200, 300, and 400 nm wide nanobars. **c** A representative image of cells cultured on nanobar substrate ($n = 8$ repeats), expressing hPiezo1-eGFP (left) and stained with CellMask Deep Red (middle). The transmitted light image is shown on the right with the width of nanobars labelled on top. **d** Averaged fluorescence images (up: hPiezo1; down: CellMask Deep Red) of nanobars with widths of 200 nm (left; $n = 141$ nanobars); 300 nm (middle; $n = 167$ nanobars); 400 nm (right; $n = 155$ nanobars). Colormap: blue: low; green: medium; red: high. In the upper right image, the dashed circles and square regions are used to calculate the 'end' and 'center'

signals, respectively, for all nanobar images. **e** Mean intensity profile along the dashed line in (**d**) for all three nanobar widths. Fluorescence traces are normalized to the intensity at the center of the nanobar. Green: hPiezo1; Magenta: CellMask Deep Red. Error bars are SEM ($n = 463$ nanobars). **f** Scattered plot of the 'end' to 'center' ratio for all nanobars in both the hPiezo1 channel and the CellMask channel (Mem.). Estimated radius of curvature for the curved ends are labelled on the x-axis. (~100 nm: $n = 141$ nanobars; ~150 nm nanobars; $n = 167$ nanobars; ~200 nm: $n = 155$ nanobars), $p$ values given by one-way ANOVA with post hoc Tukey's test. ***$p < 10^{-7}$, **$p < 10^{-3}$. All scale bars are 2 μm.

$C_0^{-1} = 83 \pm 17$ nm. The spontaneous curvature of the Piezo1-membrane complex $C_0$ represents a balance between the intrinsic curvature of Piezo1 trimers (0.04-0.2 nm$^{-1}$ as suggested by CryoEM studies[10,11,44]) and that of the associated membrane (0 nm$^{-1}$, assuming lipid bilayers alone do not have an intrinsic curvature and the formation of tether is only due to the external pulling force), consistent with the large amount of lipids associated with the dome of the propeller-shaped Piezo1 trimers[7,10]. Combining tether pulling force and tether radius measurements[23], the resulted membrane bending stiffness of Piezo1-expressing HeLa cells was $13.6 \pm 3.8$ k$_B$T ($n = 9$), comparable to that of HeLa cell blebs: $12.7 \pm 2.5$ k$_B$T ($n = 14$; Fig. S9). The similarity between the bending stiffnesses of cell and bleb membranes suggest minimal compositional change during bleb formation[46]. Using the bending stiffness of bleb membranes, we estimated the area of each Piezo1 unit $A_P = 380 \pm 170$ nm$^2$, in agreement with the area of Piezo1 trimers measured with CryoEM (~400 nm$^2$)[10,11,14,44].

## Enrichment of Piezo1 on membrane invaginations

While the previous measurements focus on the behavior of Piezo1 on membrane protrusions, the model can be directly extended to estimate the sorting of Piezo1 on membrane invaginations. It is worth noting that contributions from the size and the spontaneous curvature of Piezo1 are synergistic in the case of protrusions (Eq. 15) but can cancel each other out on invaginations (Eq. 16). Therefore, the enrichment of Piezo1 onto invaginations is predicted to be much less prominent than the depletion of Piezo1 from protrusions, with the sorting of Piezo1 peaking at 1.41 when the radius of invagination is 83 nm (Fig. S10, Methods). Moreover, the enrichment of Piezo1 is predicted to be only ($7 \pm 35$) % on 25-75 nm invaginations compared to an ($87 \pm 10$) % depletion effect on protrusions of the same range of radii. This is consistent with the lack of obvious Piezo1 enrichment spots in the bulk of the plasma membrane where highly curved (<50 nm radius) invaginations such as endocytic sites and caveolae are expected (Figs. S1–S5).

To explore potential enrichment of Piezo1 on membrane invaginations, we grew hPiezo1-expressing cells on substrates that have been finely engineered with nanobars[35,36]. Each nanobar presents two curved ends and a flat central region (Fig. 3a) and can be duplicated with high precision (Fig. 3b). When cells are cultured on top of nanobar-patterned substrates, their basal membranes will adapt to the geometry of the substrate and form membrane invaginations in parallel[36]. In hPiezo1-eGFP expressing cells, Piezo1 tends to enrich towards the curved ends, whereas the membrane marker (CellMask Deep Red) homogeneously distributes across the entire nanobar (Fig. 3c). By averaging over hundreds of nanobars with 3 different diffraction-limited widths (200, 300, 400 nm), we observed clear enrichments of Piezo1 towards the curved ends as compared to the flat central regions of nanobars (Fig. 3d, e). For all three nanobar widths, the end-to-center ratios for Piezo1 are significantly higher than those for the membrane marker (Fig. 3e, f). The median of Piezo1's enrichment at the curved ends (relative to the membrane marker) ranges from 1.31 to 1.49, comparable to the predicted sorting of Piezo1 in this range of membrane curvature (Fig. S10; $S = 1.26 - 1.40$ for invagination radius of 100 - 200 nm) and to established sensors for membrane invaginations[47]. The sorting of Piezo1 was not significantly different between the three nanobar widths (Fig. 3f), consistent with the predicted low curvature sensitivity on 100 - 200 nm invaginations ($|dS/dR| < 2$ μm$^{-1}$, which is more than 4-fold lower than $|dS/dR|$ around 55 nm protrusions; Fig. S10). While further investigations on a wider range of membrane curvature are required to fully map out the sorting of Piezo1 on membrane invaginations, our data (Fig. 3) clearly suggest that membrane curvature can lead to enrichment of Piezo1 on cellular invaginations.

## Yoda1 leads to a Ca$^{2+}$ independent increase of Piezo1 on tethers

Yoda1 is a Piezo1 agonist that has been hypothesized to bias the channel towards a less-curved state[37]. Equation (1) predicts that a smaller $C_0$ would lead to an increase of Piezo1 density on protrusions (Fig. S10). Indeed, a significantly increased amount of Piezo1 signal was observed on filopodia after adding 10 μM of Yoda1, while the radii of these filopodia remain unaltered (Fig. 4a–c, S11a). Additionally, the $S_{filo}$ of activated Piezo1 showed a positive correlation with filopodia radii (Fig. 4d; Pearson's r = 0.59). Assuming the size of Piezo1 doesn't change during activation ($\widetilde{A_P} = 2400$ nm$^2$), our model predicts a spontaneous curvature $C_0^{-1} = (4 \pm 13)$ μm of the Piezo1-membrane complex in the presence of Yoda1, corresponding to an essentially flat geometry (Fig. 4d, S10). The Yoda1 effect on Piezo1 sorting was not instantaneous, taking more than 5 min to equilibrate (Fig. S12a). This is consistent with the measured mean diffusion coefficient of Piezo1 on the cell membrane (Fig. S12b; $0.0021 \pm 0.0004$ μm$^2$/s, $n = 44$), indicating that Piezo1 trimers diffuse from the cell body to filopodia after Yoda1-induced activation.

Under regular cell culture conditions, we noticed that only a small portion of filopodia (~10%) showed measurably changed Piezo1 signal in response to the Yoda1 treatment (Fig. 4e). We reasoned that the opening of Piezo1 resembles a sharp two-state transition[14], and the main effect of Yoda1 is to lower the transition tension[37]. Piezo1 cannot be opened by Yoda1 if the resting tension of the cell is too low, whereas channels that are in a pre-stressed (yet closed) state would have a higher chance to respond to Yoda1. To test this, we pre-stressed the cells with hypotonic shock. The hypotonic shock itself did not significantly change the fraction of filopodia that showed significant sorting of Piezo1. However, significantly more (~35%) filopodia responded to a subsequent Yoda1 treatment (Figs. 4e, f and S11b, S12c). Importantly, Yoda1-induced $S_{filo}$ of Piezo1 is not a result of Ca$^{2+}$ influx, as a similar effect can be observed on cells maintained in a Ca$^{2+}$-free buffer (Fig. 4e, g). Additionally, Yoda1 induced sorting of Piezo1 is reversible as Piezo1 signals disappear from filopodia after washing out Yoda1 (Fig. 4e–g). Importantly, shear stress applied during washing

steps alone did not significantly change the sorting of Piezo1 on filopodia (Fig. S12d). On individual filopodium, the apparent all-or-none response to Yoda1 treatment suggests that cooperativity between Yoda1-Piezo1 or Piezo1-Piezo1 may also play a role in modulating the channel's curvature sorting behavior. The flattening of Piezo1 during activation has been suggested using HS-AFM[11], in silico[48,49], and has recently partially been confirmed using CryoEM[10]. Our study suggests this conformational change of Piezo1 may also happen in live cells (Fig. 4h).

## Piezo1 inhibits filopodia formation

Curvature sensing proteins often have a modulating effect on membrane geometry. For example, N-BAR proteins, which strongly enrich to positive membrane curvature, can mechanically promote endocytosis by making it easier to form membrane invaginations[40,50]. Thus, we hypothesize that Piezo1, which strongly depletes from negative membrane curvature (Figs. 1, 2), can have an inhibitory effect on the formation of membrane protrusions such as filopodia.

Indeed, HeLa cells with higher expression level of hPiezo1-eGFP tend to have less filopodia (Fig. 5a, b). However, due to the low expression of hPiezo1-eGFP in the majority of HeLa cells, only a weak negative correlation was observed between filopodia number and hPiezo1-eGFP fluorescence in each cell (Fig. 5b). In HEK293T cells where hPiezo1-eGFP expresses 3.5-fold higher, a stronger negative correlation was observed between the number of filopodia per cell and hPiezo1-eGFP density on the cell membrane (Fig. 5c, d). Interestingly, the negative correlation almost completely diminished when HEK293T cells were cultured in 5 μM Yoda1 (Fig. 5d), consistent with Yoda1's ability to reduce the curvature effects of Piezo1 (Fig. 4). In Piezo1 knockout (Piezo1-KO) cells, adding Yoda1 to the culture medium does not significantly change the number of filopodia (Fig. 5e), suggesting the agonist does not directly regulate filopodia formation without acting on Piezo1.

Next, we quantified the mechanical effect of Piezo1 by measuring the force needed to maintain 5-10 μm tethers from HEK293T cells (Fig. 5f, g). The equilibrium tether pulling force from cells overexpressing hPiezo1 was $58 \pm 18$ pN (mean ± SD, $n = 5$), significantly higher than cells without hPiezo1 overexpression ($36 \pm 6$ pN; mean ± SD, $n = 5$; Fig. 5h). Overexpressed Piezo1 in HEK293T cells (Fig. 5d–h) is ~3.5-fold higher than in HeLa cells (Fig. 5a, b), thus ~10-fold of the endogenous level (Fig. S5c). While this high Piezo1 level may resemble the local density of Piezo1 at specific cellular regions such as focal adhesion sites[27], we wonder whether the observed mechanical effect on membrane protrusions can be of general physiological relevance. To answer this question, we compared the number of filopodia on MEFs dissected from wild type (WT) or Piezo1 heterozygous (Het.) mice with those from their Piezo1-KO littermates (Fig. 5i)[25,28]. Both WT (Fig. 5j) and Het. (Fig. 5k) MEFs showed significantly less filopodia compared to their Piezo1-KO counterparts, suggesting that endogenous level Piezo1 can already inhibit filopodia formation.

## Discussion

The curved structure of Piezo1 trimer has been suggested to play an important role in the activation of the channel[7,10,11,14]. Our study demonstrates that the coupling of Piezo1 to nanoscale membrane curvature also regulates the distribution of Piezo1 within the plasma membrane. The observations of Piezo1 depletion from membrane protrusion (Figs. 1, 2) and enrichment to membrane invaginations (Fig. 3) are consistent with a recent report of Piezo1 distribution in red blood cells[19]. Our experiments also strongly support the hypothesis that Piezo1 flattens during activation[7,10,11,37], thereby coupling the activation of the channel to its subcellular distribution. It is worth noting that our data and model do not assume any molecular detail of Piezo1 and are limited by optical resolution (~500 nm). However, the revealed nano-geometries of the channel are on the order of 10 nm and are in

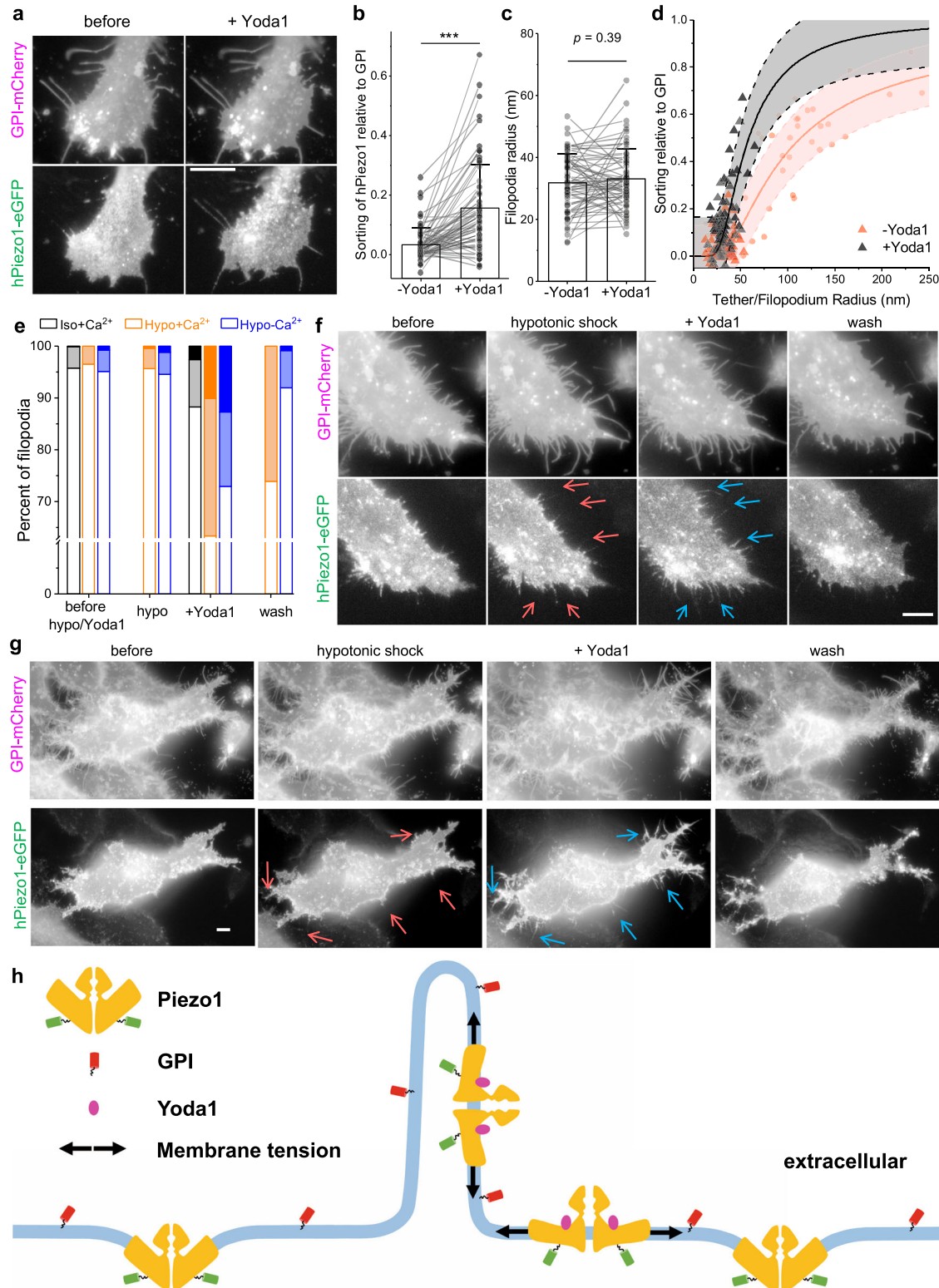

surprising agreement with the structural information of purified Piezo1 trimers[7,8,10,11].

Membrane curvature sensing has been studied for a range of proteins including BAR domain proteins and GPCRs[34,40,45,51], our report extends such quantitative studies to a mechanosensitive ion channel. Our parsimonious model predicts that the large area of the Piezo1-membrane complex (more than 10-fold larger than a typical GPCR) leads to prominent size effects that has often been neglected when studying smaller membrane proteins (Fig. S10): (1) sorting of Piezo1 is

exquisitely sensitive to the flattening of the channel ($C_0$ from 83$^{-1}$ nm$^{-1}$ to 0 nm$^{-1}$). On 50 nm protrusions, the model predicts a ~200% increase in sorting when Piezo1 flattens. This effect diminishes to ~10% if Piezo1 were 10 times smaller; (2) Piezo1 would deplete from highly curved invaginations ($S < 1$ when $R_i < 41$ nm) due to the large energy cost to match the spontaneous curvature of Piezo1 with the shape of the invaginations. The second prediction may be verified by systematically studying Piezo1 density on highly curved membrane invaginations. However, it is worth noting that we assumed a zero spontaneous

**Fig. 4 | Yoda1 leads to increased sorting of Piezo1 on filopodia, independent of Ca²⁺. a** Left: fluorescence images of a HeLa cell (full cell: Fig. S11a) co-expressing GPI-mCherry (up) and hPiezo1-eGFP (down). Right: 10 min after adding 10 μM Yoda1. **b, c** Quantifications of hPiezo1 sorting on filopodia (**b**) and filopodia radii (**c**), $n = 66$ filopodia. Bar plots show mean + SD. $p$ values given by 2-tailed paired Student's $t$ test, ***$p < 10^{-7}$. **d** $S_{filo}$ plotted as a function of filopodia radius before (red triangle) and after (black triangle) adding Yoda1. The center and error band represent the best fit of $S_{filo}$ (+Yoda1; $n = 66$ filopodia) to Eq. (1) ($\widetilde{A_P}$ fixed at 2400 nm²) and the 90% confidence interval, respectively. Fitted $C_O^{-1} = (4 \pm 13)$ μm. Data from Fig. 2g shown in pink. All data points in (**b**)–(**d**) are quantified from cells cultured in regular XC and are limited to filopodia that did not significantly change positions after addition of Yoda1. **e** Percentage of filopodia that showed strong ($S_{filo} > 0.3$, dark), weak ($0.1 < S_{filo} < 0.3$, light), and no ($S_{filo} < 0.1$, open) sorting of hPiezo1 under the labelled conditions. Black: no osmotic shock, regular XC ($n = 752$ filopodia, as in **a**); Orange: hypotonic shock, regular XC ($n = 564$ filopodia, as in **f**); blue: hypotonic shock, Ca²⁺-free XC ($n = 771$ filopodia, as in **g**). **f, g** Fluorescence images of HeLa cells in regular (**f**, full cell shown in Fig. S11b) and Ca²⁺-free (**g**) XC buffer. Up: GPI-mCherry. Down: hPiezo1-eGFP. Left to right: before treatments; 10 min after swelling with regular (**f**) or Ca²⁺ free (**g**) hypotonic buffer; 20 min after adding 10 μM Yoda1 (dissolved in regular (**f**) or Ca²⁺ free (**g**) hypotonic buffer); after washing 3 times with regular (**f**) or Ca²⁺ free (**g**) XC buffer. Red/blue arrows point to the filopodia where Piezo1 signals were absent/enhanced before/after adding Yoda1. All fluorescence images are shown in log-scale to highlight the filopodia. All scale bars are 5 μm. **h** Illustration showing the membrane curvature sorting of Piezo1 relative to GPI and the effect of Yoda1 and pre-stressing (arrows) on the curvature sorting of Piezo1.

curvature for membranes associated with Piezo1 and that the spontaneous curvature of the Piezo1-membrane complex is independent of the shape of surrounding membranes. These assumptions may no longer hold when studying Piezo1 in highly curved invaginations or liposomes[11]. Furthermore, our model assumes that Piezo1 behaves as 2-dimensional ideal gas in the membrane. While the assumption is consistent with the observation that Piezo1 trimers function independently[52], we did neglect potentially important intermolecular interactions between Piezo1 trimers[25,28,49,53]. Notably, apparent clusters of Piezo1 were often observed on cell membranes (Fig. 1, Fig. S1–S5) and on membrane tethers (Fig. 2d), suggesting that attractions between Piezo1 trimers may also be important for controlling the detailed distribution of the channel in cells. Lastly, Piezo1-membrane complexes likely have isotropic curvatures (such as a spherical dome[7]), whereas the membrane protrusions and invaginations in our study are anisotropically curved, with one principal membrane curvature close to zero. Improved models that consider the effect of curvature anisotropy may provide additional insights to the curvature sensing of Piezo1[54,55].

In addition to membrane curvature, tension in the membrane may also affect the subcellular distribution of Piezo1[21]. Particularly, membrane tension can activate the channel and potentially change Piezo1's nano-geometries. This tension effect is unlikely to play a significant role in our interpretation of the curvature sorting of Piezo1 (Fig. 2): (1) HeLa cell membrane tension as probed by short tethers (Fig. S9f; $45 \pm 29$ pN/μm on blebs and $270 \pm 29$ pN/μm on cells, with the highest recorded tension at 426 pN/μm) are significantly lower than the activation tension for Piezo1 (>1000 pN/μm[12,13,23,56]). (2) With more activated (and potentially flat) channels under high membrane tension, one would expect a higher density of Piezo1 on tethers pulled from tenser blebs. This is the opposite to our observations in Fig. 2c–g, where Piezo1 density on tethers was found to decrease with the absolute curvature, thus tension (Eq. 7), of membrane tethers.

A Yoda1-induced flattening of Piezo1 has not been directly observed via CryoEM. Our results (Fig. 4) point towards two challenges in determining this potential structural change: (1) Yoda1 induced changes in Piezo1 sorting is greatly amplified after pre-stretching the membrane (Fig. 4e), pointing to the possibility that a significant tension in the membrane is required for the flattening of Yoda1-bound Piezo1. (2) Piezo1 is often incorporated in small (<20 nm radius) liposomes for CryoEM studies. The shape of liposomes can confine the nano-geometry of Piezo1[10,11], rendering it significantly more challenging to respond to potential Yoda1 effects. This potential influence of membrane curvature on the activation of Piezo1 would be an interesting direction for future studies.

The antagonistic effect of Piezo1 on the formation of filopodia is consistent with several recent observations: First, Piezo can modulate the morphogenesis and targeting of dendrites independently of its activity as a mechanosensitive ion channel[57]. Secondly, knocking out Piezo in Drosophila promotes axon regeneration[58]. Furthermore, Piezo1 negatively regulates the morphological activity (i.e., number of protrusions) of muscle stem cells[59]. Lastly, Piezo2 inhibits neurite outgrowth in N2A cells[17]. In addition to membrane curvature, several parallel Piezo-related mechanisms can regulate the formation and growth of filopodia, including Ca²⁺ signaling induced by the activation of Piezo and potential interactions between specific Piezo domains with cytoskeletal components[17,58,59]. Our data do not exclude these parallel mechanisms, rather, we suggest that the curvature preference of Piezo1 provides an additional route to control filopodia dynamics. Further studies are required to fully dissect the contribution of each of these variables.

In addition to regulating the formation of filopodia (Fig. 4), the curvature sensing of Piezo1 can have a direct benefit of making sure that the protein is retained at the rear edge during cell migration and to avoid losing Piezo1 to retraction fibers (e.g., Fig. S4b)[28]. Moreover, the dynamics of filopodia are often linked to the metastatic transition of cancer cells[41], suggesting new roles of Piezo1 in cancer biology[29].

Overall, our study suggests that the curvature sensing of Piezo1 provides nanoscale input for the channel in live cells and is a universal regulator of the channel's distribution within cell membranes. These features are likely to be of fundamental importance to a wide range of Piezo1-dependent biological processes.

## Methods
### Cell harvesting and culture, transfection, bleb formation, osmotic shock, and Yoda1 treatment
HeLa cells (ATCC and from Renping Zhou lab, Rutgers) were cultured in Eagle's Minimum Essential Medium (EMEM) supplemented with 10% Fetal Bovine Serum (FBS) and 1% Penicillin/Streptomycin (PS). HEK293T cells (ATCC and from Zhiping Pang lab, Rutgers) were cultured in Dulbecco's Modified Eagle Medium (DMEM) supplemented with 10% FBS, 1% PS and 1% Sodium Pyruvate. Homosapiens bone osteosarcoma U2OS cells (ATCC) were maintained in DMEM with GlutaMAX (Gibco) supplemented medium with 10% FBS and 1% PS. All cell lines were seeded in 100 mm plastic dishes with ~$1 \times 10^6$ cells per dish. Cells were kept in incubator with 5% CO₂ and 100% relative humidity at 37 °C.

All animal studies were approved by the Institutional Animal Care and Use Committee of University of California at Irvine and performed in accordance with their guidelines. To obtain Mouse Embryonic Fibroblasts (MEFs), mice that are heterozygous for Piezo1 knockout (Piezo1$^{\Delta/+}$) were obtained from Jackson Laboratories (JAX stock 026948) and were bred with C57BL6/J to maintain the colony. Piezo1$^{\Delta/+}$ mice were bred with each other to generate a mixture of wild type (WT), heterozygous (Het.), and knockout (Piezo1-KO) embryos. Mice were considered embryonic day 0.5 upon vaginal plugging. Embryos were genotyped via a commercial vendor (Transnetyx). MEFs with endogenously labelled Piezo1 were harvested from Piezo1-tdTomato reporter mice (*Mus. musculus*, JAX stock 029214), expressing Piezo1 with a tdTomato knock-in at the C-terminus (gift from Patapoutian lab)[60].

Embryos were dissected by separating the head, limbs, and tail from the embryo at embryonic days 10.5 (WT, Het, Piezo1-KO) or 12.5

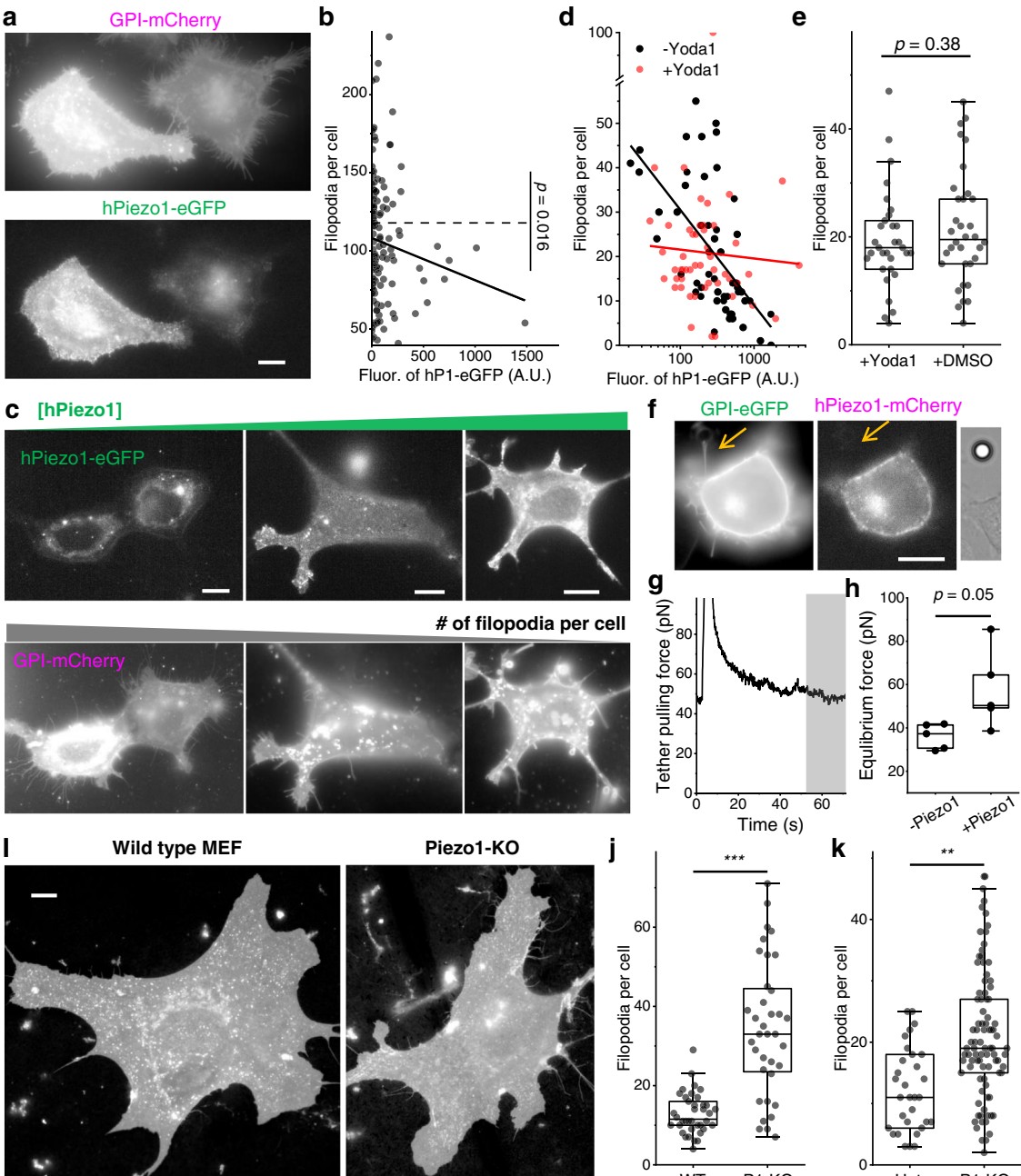

**Fig. 5 | Piezo1 inhibits filopodia formation. a** Fluorescence images of HeLa cells co-expressing GPI-mCherry (up) and hPiezo1-eGFP (down). **b** Relation between the number of filopodia and hPiezo1-eGFP expression level in HeLa cells ($n = 129$ cells). Dash line: average. Solid line: linear fit (Pearson's r = −0.13). **c** Fluorescence images of HEK293T cells co-expressing hPiezo1-eGFP (up) and GPI-mCherry (down). **d** Relation between the number of filopodia and hPiezo1-eGFP expression level in HEK293T cells cultured in regular (black, $n = 52$ cells) and 5 μM Yoda1 containing (red, $n = 50$ cells) media. Lines are linear fits between y and log(x). Without Yoda1 (black): slope = −21.5 ± 4.0, Pearson's r = −0.61. With Yoda1 (red): slope = −2.0 ± 5.0, Pearson's r = −0.06. **e** The number of filopodia per Piezo1-KO MEF incubated with 5 μM Yoda1 ($n = 31$ cells) or 0.1% DMSO ($n = 34$ cells). **f** A HEK293T cell co-expressing hPiezo1-eGFP (left) and GPI-mCherry (middle), with a tether (arrow) pulled by an optically-trapped bead (right). **g** Time dependent pulling force for the tether in (**f**).

The tether was stretched at $t = 3$ s. The gray area was used to calculate the equilibrated pulling force. **h** Equilibrium tether pulling force for HEK293T cells only expressing GPI-eGFP (−Piezo1; $n = 5$ cells) or co-expressing hPiezo1-mCherry and GPI-eGFP (+Piezo1; $n = 5$ cells). **i** Representative (of the median filopodia number) fluorescence images of WT (left) and Piezo1-KO (right) MEF, both stained with CellMask Deep Red. MEFs were from littermate embryos dissected on the same day. **j** The number of filopodia per cell for WT ($n = 38$ cells) and Piezo1-KO (P1-KO; $n = 36$ cells) MEFs from a pair of littermates. Cells were imaged 30 days after dissection. **k** The number of filopodia per cell for Het. ($n = 32$ cells) and P1-KO ($n = 89$ cells) MEFs from a pair of littermates. Cells were imaged 52 days after dissection. All fluorescence images are shown in log-scale to highlight the filopodia. All scale bars are 10 μm. $p$ values given by two-tailed Student's $t$ test after checking for normality. ***$p < 10^{-7}$, **$p < 10^{-3}$.

(Piezo1-tdTomato) in Dulbecco's PBS (Gibco) supplemented with 33 mM D-(+)-glucose (Sigma-Aldrich) and 1% Penicillin-Streptomycin (10,000 U/mL; Gibco). Following dissection, tissue was spun down at ~260 G for 5 min, and the supernatant was aspirated. The remaining tissue from each embryo was gently triturated in DMEM

(ThermoFisher Scientific) with 15% FBS (Omega Scientific), 1x Gluta-Max (ThermoFisher Scientific), 1 mM sodium pyruvate (ThermoFisher Scientific), and 1x non-essential amino acid solution (ThermoFisher Scientific) in a sterile environment to obtain single cells. All MEFs were grown in a 5% $CO_2$ incubator at 37 °C, and media was changed 24 H

after plating. Cells from individual embryos obtained from the Het. crosses were cultured separately in 12-well plates (USA Scientific) coated with 0.1% gelatin solution (Fisher Scientific) until genotypes were determined. Subsequently, cells were pooled together by genotype and cultured. Cells were grown until 90% confluency was reached. All MEFs were passaged using TrypLE Express (ThermoFisher) to dissociate the cells and were spun at ~260 G for 5 min. Cells were passaged at least three times prior to use in experiments.

For fluorescence imaging, cells were transfected in 35 mm plastic dishes at around 60% confluency. For HeLa cells, a mixture of 125 μL Opti-MEM, 3.5 μL TransIT-X2 and 300 ng of each plasmid DNA were added after 12 min incubation. For MEFs and HEK293T cells, a mixture of 250 μL Opti-MEM, 5 μL P3000 reagent, 3.5 μL Lipofectamine reagent and 500 ng of each plasmid DNA were added after 15 min incubation. Cells were kept in an incubator for 24 H before further split onto 35 mm precoated (Poly-D-Lysine for HeLa cells and Matrigel for MEFs and HEK293T cells) glass bottom dishes (Cellvis) for imaging. Cells were imaged 12-48 H after splitting.

Before imaging, cell culture medium was replaced with extracellular imaging buffer (XC buffer) containing 125 mM NaCl, 2.5 mM KCl, 15 mM HEPES, 30 mM Glucose, 1 mM MgCl₂ and 3 mM CaCl₂ with pH 7.3. $Ca^{2+}$ free experiments were done by omitting CaCl₂ in the XC buffer. Hypotonic osmotic shock was carried out by using a diluted XC buffer that is 0.25-0.5 of its initial concentration.

To trigger blebs, HeLa cells were incubated with 400 μL of 100-200 μM Latrunculin B (diluted in XC buffer) for 5 min. Additional XC buffer was added after bleb formation to keep a final Latrunculin concentration of 20-40 μM during experiments.

For experiments in Fig. 5, stock solution of Yoda1 (5 mM, dissolved in DMSO) was diluted to 5 μM in EMEM before use. Cells were incubated with 5 μM Yoda1 solution for 2 to 4 H. Control group was incubated with 0.1% DMSO for the same amount of time. After incubation, culture media were carefully replaced with XC buffer containing the same concentrations of Yoda1 or DMSO. For experiments in Fig. 4, stock solution of Yoda1 was diluted to 10 μM in XC buffer before use. The number of filopodia per cell in Fig. 5 was counted manually for all protrusions that are longer than 1 μm for HEK293T cells and longer than 2 μm for MEFs and HeLa cells, counting was independently verified by three researchers and by FiloDetect[61].

## Imaging, tether pulling, and quantification

Fluorescence imaging was done on either a Leica DMi8 or a Nikon Ti2-A inverted microscope. Leica DMi8 was equipped with an oil-immersion objective (100X; NA 1.47; Leica) and laser excitation (488 nm for eGFP and 561 nm for mCherry, tdTomato, or mOrange2), and allows total internal reflection fluorescence (TIRF) imaging. Nikon Ti2-A microscope was equipped with LED excitation (~470 nm for eGFP and ~550 nm for mCherry, tdTomato, or mOrange2) and with either a water-immersion objective (60X; NA 1.20; Nikon) or an oil-immersion objective (100X; NA 1.30; Nikon). The oil immersion objective was integrated with an objective heater (OKO lab) for 37 °C measurements. Temperature was calibrated by directly measuring the temperature of the medium near the imaged cell. The Leica DMi8 was integrated with an Infinity Scanner (Leica) for fluorescent recovery after photobleaching (FRAP) experiments. The Nikon Ti2-A microscope was integrated with two micromanipulators (PatchPro-5000, Scientifica) and an optical tweezer (Tweez305, Aresis) for tether pulling and force measurements. Images were analyzed with ImageJ and Matlab (R2019a).

Membrane tethers were pulled from the cell body or cell-attached membrane blebs using either a motorized micropipette[62] with a fused tip or a polystyrene bead (4.5 μm) trapped with an optical tweezer.

Membrane curvature sorting and filopodia/tether radii were calculated according to the illustration in Fig. S3. We hypothesize that filopodia and membrane tether are cylindrical membrane tubes, and

that the quantum yields of fluorescent proteins are independent of local curvature. For radius calculation, we also assume that both bilayers of a flat region of the cell are captured in widefield epifluorescence images[23].

For membrane tube with a diffraction-limited radius $r$, if a region of interest (ROI) is drawn to cover a length $L$ of the tube (Fig. S3b), the total membrane area with the ROI is given by

$$A_{tube} = 2\pi r L \tag{2}$$

Assume $I_{mean}^{tube}$ is the background-corrected mean fluorescence within the ROI on the tube, $A_{ROI}$ is the area of the ROI, the total fluorescence should equal to the number density of FPs on membrane ($\rho$) multiplied by $A_{tube}$ and by the fluorescence per FP ($\beta$).

$$I_{mean}^{tube} \cdot A_{ROI} = A_{tube} \cdot \rho \cdot \beta \tag{3}$$

The background-corrected mean fluorescence within the ROI on a flat region of the cell membrane (Fig. S3b), $I_{mean}^{cell}$, is related to $\rho$ and $\beta$ by

$$I_{mean}^{cell} = M \cdot \rho \cdot \beta \tag{4}$$

The factor $M$ takes into account the number of cell membrane surfaces within the imaging depth.

Combining Eq. 2 ~ Eq. 4, the radius of the membrane tube is given by

$$r = \frac{M \cdot I_{mean}^{tube} \cdot A_{ROI}}{2\pi L \cdot I_{mean}^{cell}} \tag{5}$$

When imaging a flat region of the cell under widefield fluorescence microscopy, we use $M = 2$ (Fig. S3d). HEK293T cells often do not contain a flat region on the cell body (Figs. S2e, S4). To address this issue, we use TIRF microscopy and set $M = 1$ when measuring filopodia radii from HEK293T cells.

In principle, $r$ can be calculated using the fluorescence of any membrane proteins/lipids, however, only molecules in the membrane that do not have membrane curvature sensitivity can give the real tube radius. The reported tether and filopodia radii in our study were determined using the fluorescence of GPI-FP. GPI anchors the FP to the outer leaflet of the plasma membrane. Due to the relatively large size of FP to the GPI anchor, GPI-FPs may have enrichment towards highly curved protrusions[63]. For the same reason, CaaX-FP, which is anchored to the inner leaflet of the plasma membrane may deplete from membrane protrusions. However, we expect this effect to be less than 7%, as the relative sorting of CaaX to GPI was measured to be 0.863 ± 0.008 (Fig. 1d).

When a tether is pulled from a bleb (Fig. 3c, d), an apparent radius of the tether was calculated using the background-corrected mean fluorescence within the ROI of the bleb membrane instead of $I_{mean}^{cell}$ (Fig. S3c).

Membrane curvature sorting ($S$) on a protrusion is the effective number density of a molecule of interest (MOI) on the membrane tube relative to either a flat region on the cell body (Fig. S3b) or the membrane bleb (Fig. S3c).

$$S = \frac{r(MOI)}{r(GPI)} \tag{6}$$

$S = 0$ when the MOI is completely depleted from the tube. $S = 1$ when the MOI has the same curvature preference as the GPI-FP reference.

An ROI width of 0.5 μm was used for all tether/filopodium quantifications. Filopodia that are sufficiently away (>1 μm) from other membrane structures and more than 3 μm in length were picked for

calculating $S$ and $r$. Errors in $S$ and $r$ were propagated using the standard deviation in the background fluorescence as the error for mean fluorescent intensities. In Fig. 4e, filopodia with clear Piezo1 signal on less than 10% of the total length (typically corresponding to $S < 0.1$) were considered as no response; between 10% to 50% of the filopodia (typically corresponding to $0.1 < S < 0.3$) were considered weak response; more than 50% of the filopodia ($S > 0.3$) were considered strong response.

### Fabrication, cell culture, imaging, analysis with nanobar-substrates

Nanobar substrates were fabricated on the surface of a square quartz coverslip with electron-beam lithography (EBL)[35,36]. Briefly, the quartz chip was firstly spin-coated with the poly(methyl methacrylate) (PMMA) (MicroChem) with around 300 nm height, followed by one layer of conductive polymer AR-PC 5090.02 (Allresist). Nanobar patterns were written by Electron Beam Lithography (FEI Helios NanoLab), and then developed in isopropanol: methylisobutylketone solution with 3:1 ratio. After that, a 30 nm thickness chromium (Cr) mask was deposited by thermal evaporation (UNIVEX 250 Benchtop) and then lifted off with acetone. Nanobars were then generated through reactive ion etching (RIE) with a mixture of CHF3 and CF4 (Oxford Plasmalab). Nanobar dimensions were identified by scanning Electron Microscopy (FEI Helios NanoLab) imaging with 10 nm Cr coating.

To enable cell imaging on nanochip, the chip was attached to the 35 mm cell culture dish (TPP) with a hole punched in the center to expose nanobar pattern. Before cell plating, the dish substrate was sterilized by UV treatment for 20 min and treated with high-power air plasma (Harrick Plasma) for 3 min. The surface was coated with 0.2% Gelatin (Sigma-Aldrich) for 30 min to promote cell attachment. U2OS cells were then cultured on the nanobar chip for one day before transfection. Cells were transfected with hPiezo1-eGFP plasmid via Lipofectamine. Membrane staining with CellMask Deep Red (Life Technologies) was performed before imaging. The 1000-time diluted dye was added to the cells and incubated for 1 min at 37 °C and washed with DMEM.

Cell imaging was performed with a spinning disc confocal (SDC) built around a Nikon Ti2 inverted microscope containing a Yokogawa CSU-W1 confocal spinning head and a 100 X/1.4NA oil immersion objective. eGFP was excited at 488 nm and detected at 505–545 nm. CellMask Deep Red was excited at 639 nm and detected at 672–712 nm. During imaging, the cells were maintained under 37 °C with 5% CO2 in an on-stage incubator.

The Piezo1-eGFP curvature sensing preference on gradient nanobars with different widths was quantified using Matlab[35,36]. In brief, the background signals of each image were firstly subtracted by the rolling ball algorithm in ImageJ (radius = 0.77 μm). Next, the nanobars covered by cell membrane were identified by square masks (3.92 μm × 3.92 μm) centered at the nanobar. The average images were generated by averaging all the individual nanobar masks. To quantify the end-to center ratio, each nanobar was segmented into three ROIs (two nanobar-ends and a nanobar-center; Fig. 3d). End-to-center ratios were calculated by dividing the mean end-intensity with the mean center-intensity. The curvature sorting of Piezo1 on membrane invaginations is evaluated by dividing the end-to-center ratio in the Piezo1 channel by that of the CellMask channel.

### Tether force measurement

To measure tether force, a membrane tether is pulled by a bead trapped with an optical tweezer (Tweez305, Aresis) equipped on the Ti2-A inverted microscope (Nikon). Membrane tubes were pulled to around 10 μm in length and then held until an apparent equilibrium force $f$ was reached (Fig. S9). Then fluorescence images of the cell and the tether were taken for tether radius ($r$) measurements according to Eq. 5. Force on the bead was calculated from the displacement of the

bead from the center of the trap and the trap stiffness (calibrated before each experiment by applying equipartition theorem to the thermal fluctuation of a trapped bead). Then membrane tension $\sigma$ and bending stiffness $\kappa_m$ were be calculated by[43]:

$$\sigma = \frac{f}{2\pi r} \tag{7}$$

$$\kappa_{\mathrm{m}} = \frac{fr}{2\pi} \tag{8}$$

Note that the pulling force $f$ may contain contributions from the cytoskeleton and membrane asymmetry. Therefore, a more accurate measure of $\kappa_m$ is to fit $f/2\pi$ vs. $r^{-1}$ to a linear relation where the slope will report $\kappa_m$ and the intercept will report the aforementioned additional contributions to tether pulling force (Fig. S9f).

### FRAP measurements

3 to 6 circular ROIs with radius $R_{\mathrm{bleach}} = 1.5\,\mu m$ were picked on flat regions of HeLa cell expressing hPiezo1-eGFP. 488 nm laser at full power was used to photo bleach the selected ROI for -1 s. Frames before photobleaching and the first frame after photobleaching were used to normalize the fluorescence intensity, background photobleaching was corrected by tracking the fluorescence of the entire cell. The normalized recovery curve $I$(t) was fitted to the following relation to extract half-recovery time ($\tau_{0.5}$, 2-parameter fit):

$$I(t) = \frac{I_0 + \frac{t}{\tau_{0.5}}}{1 + \frac{t}{\tau_{0.5}}} \tag{9}$$

The diffusion coefficient was calculated by[23,64]:

$$D_{\mathrm{cell}} = 0.224 \frac{R_{\mathrm{bleach}}^2}{\tau_{0.5}} \tag{10}$$

### Model for the curvature sorting of Piezo1

We assume the protein-membrane complex has a preferred curvature $C_0$. In the case of Piezo1, each protein unit would correspond to a trimer of Piezo1 with associated membranes, $C_0$ represents a balanced shape between the curvature preference of the Piezo1 trimer and the preferred shape of the associated membrane. If we define protrusions (e.g., tethers, filopodia) on the cell to have a negative curvature, then $C_0$ would be predicted to be positive for Piezo1-membrane complex, assuming that the associated membranes prefer to be flat.

The energy of putting one unit of protein-membrane complex into a membrane of curvature $K$:

$$E^{\mathrm{b}} = \frac{1}{2}\widetilde{\kappa}A_{\mathrm{P}}(K - C_0)^2 \tag{11}$$

$K$ is the sum of principle curvatures of the membrane surface. For a flat membrane, $K = 0$; for tubular membrane protrusions of radius $R_{\mathrm{p}}$ (e.g., filopodia, tether), $K = -1/R_{\mathrm{p}}$; for tubular membrane invaginations of radius $R_{\mathrm{i}}$, $K = 1/R_{\mathrm{i}}$. Here, we ignored the contribution of Gaussian curvature (the product of principle curvatures) which is expected to be a constant that only depends on the boundary conditions (Gauss-Bonnet theorem)[65]. $A_{\mathrm{p}}$ is the area of the protein-membrane complex. $\widetilde{\kappa}$ is the bending stiffness of the protein-membrane complex, which represents the stiffness when bending the membrane ($\kappa_{\mathrm{m}}$) and the protein ($\kappa_{\mathrm{p}}$) in series:

$$\frac{1}{\widetilde{\kappa}} = \frac{1 - \theta_{\mathrm{p}}}{\kappa_{\mathrm{m}}} + \frac{\theta_{\mathrm{p}}}{\kappa_{\mathrm{p}}} = \frac{1}{\kappa_{\mathrm{m}}} + \theta_{\mathrm{p}}\left(\frac{1}{\kappa_{\mathrm{p}}} - \frac{1}{\kappa_{\mathrm{m}}}\right) \tag{12}$$

Here, $\theta_p$ is the area fraction of the protein in each unit of protein-membrane complex. Measurements of the overall membrane bending stiffness of Piezo1 expressing cells ($\sim \kappa_m$), therefore serve as a good estimation for $\tilde{\kappa}$ when either $\theta_p$ is small or when $\kappa_p \approx \kappa_m$. A recent study suggests that Piezo1 is surprisingly flexible, presenting a bending stiffness similar to that of the lipid membrane[66].

The energy difference between a protein on the curved membrane versus the same protein on a flat membrane:

$$\triangle E = E^b(\text{curved}) - E^b(\text{flat}) = \frac{1}{2}\tilde{\kappa}A_P(K-C_0)^2 - \frac{1}{2}\tilde{\kappa}A_P(C_0)^2 = \frac{1}{2}\tilde{\kappa}A_P(K^2 - 2C_0 K)$$

(13)

$\Delta E$ describes the energy change of moving a protein-membrane complex from a flat membrane to a curved membrane, where the flat-curve membrane geometry was pre-equilibrated (e.g., a tube pulled from a piece of flat membrane and is held by an external force). Note that the model predicts an energy cost of moving a 'flat protein' ($C_0 = 0$) from a flat to a curved region, this is because the 'flat protein' will deform the pre-equilibrated curvature, with an energy cost that is larger for bigger proteins. The reference molecule GPI has a much smaller area in the membrane ($\sim 1 \, nm^2$) compared to Piezo1, therefore $\Delta E$ for GPI is negligible.

Assume the proteins in the membrane can be approximated as 2D ideal gas (i.e., no interaction between proteins, density of protein on the membrane is low). The density of the protein on the curved membrane relative to its density on the flat membrane follows the Boltzmann distribution:

$$S = \exp\left(\frac{-\triangle E}{k_B T}\right) = \exp\left[-\frac{\tilde{\kappa}A_P}{2k_B T}\left(K^2 - 2C_0 K\right)\right]$$

(14)

Note that Eq. 14 is essentially the same as Eq. 8 of ref. 34 or Eq. 2 of ref. 45 (under low protein density and negligible protein-protein interaction), where free energy-based derivations were presented. For membrane protrusions of radius $R_p$, $K = -1/R_p$:

$$S = \exp\left[-\frac{\tilde{\kappa}A_P}{2k_B T}\left(\frac{1}{R_p^2} + \frac{2C_0}{R_p}\right)\right]$$

(15)

which was used to fit the sorting of Piezo1 on membrane tethers (Fig. 2g; Eq. 1) using OriginPro (2020). The resulted fitting parameters were, $\tilde{\kappa}A_P = 4800 \, k_B T \cdot nm^2$ and $C_0^{-1} = 83 \, nm$, corresponding to the nano-geometry of Piezo1-membrane complex under a resting state. Fix $\tilde{\kappa}A_P = 4800 \, k_B T \cdot nm^2$ and fit the data in Fig. 3d (+Yoda1) to Eq. 15, we get $C_0 = 0 \, nm^{-1}$, potentially corresponding to the nano-geometry of Piezo1-membrane complex under open or inactive state.

For membrane invaginations of radius $R_i$, $K = 1/R_i$:

$$S = \exp\left[-\frac{\tilde{\kappa}A_P}{2k_B T}\left(\frac{1}{R_i^2} - \frac{2C_0}{R_i}\right)\right]$$

(16)

Note that in Eq. 15, $S$ is a monotonic increasing function of protrusion radius $R_p$. However, the sorting of Piezo1 on membrane invaginations (Eq. 16) peaks at invagination radius $R_i = 1/C_0$ (Fig. S10). Fig. S10 also shows that the effect of channel opening on Piezo1 sorting (changing $C_0$ from $83^{-1} \, nm^{-1}$ to $0 \, nm^{-1}$) would almost diminish if Piezo1 were 10 times smaller. Lastly, curvature sensing of the 10-time smaller protein on membrane invaginations would be significantly stronger if the protein also has a 10-time larger spontaneous curvature $C_0$ ($8.3^{-1} \, nm^{-1}$), similar to those of typical BAR domain proteins[40,45].

## Reporting summary

Further information on research design is available in the Nature Portfolio Reporting Summary linked to this article.

## Data availability

Source data are provided with this paper.

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

## Acknowledgements

The authors are grateful for a critical review of a preprint version of this manuscript (https://sciety.org/articles/activity/10.1101/2022.06.22.497259) by Alec Nickolls, Ruhma Syeda, Bailong Xiao, and Alex Chesler. We thank Bailong Xiao lab for providing quantifications of CryoEM images of Piezo1 containing liposomes. We thank Gill Fitz and Matt Tyska for helpful suggestions on filopodia quantification. We thank Bineet Sharma and Joyce Lin for a careful proofread of the manuscript. We thank Ben Schuster lab for access to confocal microscopy, Renping Zhou, Zhiping Pang, and Tibor Rohacs labs for helping with cell culture. We thank Markus Deserno, Padmini Rangamani, Bianxiao Cui, Elizabeth Kelley, Navid Bavi, Rod MacKinnon, Liqun Luo, Jiefu Li, Jie Xu, Andy Nieuwkoop, Deirdre O'Carroll, and Rick Remsing for helpful discussions. The project is supported by the National Institute of General Medical Sciences of the National Institutes of Health (NIH) under Award Number R35GM147027 (Z.S.) and by the National Institute on Drug Abuse, the National Institute of

Neurological Disorders and Stroke, and the National Institute of Mental Health of the NIH under Award Number R21DA056322 (Z.S.). H.W. is supported by the Zhou family fellowship. A.T.L. acknowledges the R01(NS10981) Diversity Supplement and NIH F31 1F31NS127594-0. M.M.P acknowledges R01NS109810. W.Z. acknowledges the funding supports from the Singapore Ministry of Education (MOE) RG95/21 and the Human Frontier Science Program Foundation RGY0088/2021.

## Author contributions

Z.S. and C.D.C. conceived the project. S.Y. carried out the majority of the experiments and analyzed the data. X.M., X.G., and W.Z. designed and carried out the measurements in Fig. 3. S.A. carried out parts of the measurements in Fig. 4 and Fig. S12. B.L. carried out parts of the measurements in Fig. 5, Fig. S4, and Fig. S9. A.T.L. and M.M.P. designed and prepared Piezo1-KO, Piezo1-heterozygous, Piezo1-tdTomato, and wild type MEFs. H.W. helped with confocal imaging and the preparations of HEK293T cells and plasmids. M.W. helped with the imaging of D2R expressing cells and the theoretical modeling. Z.S. oversaw the experiments and data analysis of the entire project. Z.S., S.Y., and C.D.C. wrote the manuscript with input from all authors.

## Competing interests

The authors declare no competing interests.
