## [Peer Review File · Nature Communications]

REVIEWER COMMENTS

Reviewer #1 (Remarks to the Author):

The MS of Yang et al. deals with depletion of human/mouse PIEZO1 from cellular protrusions (filopodia, and artificial tethers pulled directly from the plasma membrane/PM or from cell-attached Giant PM Vesicles/blebs). The depletion is always relative to membrane bound anchors (Caax, GPI) and is attributed to the preferential sorting of hPIEZO1 away from protrusions of high positive membrane curvature (+C). Yoda1 and hypo-osmotic pressure appear to regulate depletion suggesting the effect depends on the conformation of the mechanochannel. hPIEZO1 expression downregulates filopodia expression attributing curvature induction properties to PIEZO1.

This is a very interesting manuscript that is in principle relevant for NatComm because it aims to connect for the first time the remarkable deformations PIEZO1 induces to membranes, with tension (the classical functional regulator) and its sorting behaviour. However, in my opinion, unfortunately, there are a number of major/critical issues that have to be addressed before the main claims are validated and thus before considering publication.

1. PM Distribution \neq organelle specific depletion. This MS deals exclusively with PIEZO1 1) depletion in 2) filopodia/protrusions. However, these two “corner stones of the findings” are entirely absent from the title, the abstract and the discussion. The sentence “regulator of distribution to the PM of living cells” does not appear to be justified by the data which show sorting from/to a specific “organelle” i.e. filopodia. The term “distribution”, which typically refers to “upconcentration”, appears also inappropriate for data that only show “varying degrees of depletion”. In light of the above, the authors may wish to consider focusing/emphasising in the discussion the potential biological role of PIEZO1 depletion from filopodia.

2. Fig. 1 aims to demonstrate that average PIEZO1 filopodia densities are smaller than GPI, CaaX, D2R, TREK-1. However, if protein overexpression affects filopodia size distribution this comparison is meaningless (since densities are hypothesised to be curvature depended). This does not appear to be a problem for PIEZO1 but seems to be a problem e.g. for D2R. How about TREK? Please add in 1E filopodia size distributions for all used constructs. PS, I would have shown a zoom of the same area from 1A in the two channels to highlight there is no PIEZO1 signal.

3. Fig. 2 aims to establish that depletion scales with tether radius. 2E shows that 1) bleb-tethers from floppy (low tension) blebs have 2) larger diameters and 3) smaller depletion than other bleb-tethers (cell-tethers and filopodia). It is not clear to me why/how the authors deconvolve the possible influence of tension from curvature and conclude that only the latter is important (see also line 316).

Regarding the model:

4. This is very simple phenomenological model. It is perfectly fine to use phenomenological models to fit experimental data, because obtaining a good fit (for reasonable parameters) validates the model. However, in my opinion, such models should not be used to extrapolate to areas where no data have ever been collected (negative curvature in this case, S8) and where therefore no validation can be obtained. Especially for such complex giga-Dalton proteins as the PIEZO, which has furthermore an inherently asymmetric structure and thus likely asymmetric behavior as a function of curvature.
5. There appears to be a crucial misunderstanding. The spontaneous radius of curvature that comes out of the model is the curvature at which the membrane-protein-mismatch potential energy would be minimized. This is equal to the actual “nano-geometry in situ” of protein at the PM only! if this protein dominates over all competing membrane deformation/stabilisation processes. This however cannot be assumed, it rather has to be proved. In my opinion/understanding the data presented do not resolve nor provide any info regarding the actual membrane geometries around PIEZO1.
6. If I understood correctly the model provided a spontaneous radius of curvature of -87 nm. This appears to be an order of magnitude larger than the ~10nm obtained by CryoEM, however the author suggests the numbers are in good agreement (line 205). Sorry if I misunderstood something...?
7. The model is using bending stiffness of cellular membranes however, it is fitting bleb data. Isn't more appropriate to measure directly the bending stiffness of blebs?

Figure 3

8. Fig. 3 aims to demonstrate that yoda1 application which activates the channel, changes its coupling to curvature. This is a crucial figure in the manuscript, in my opinion the most important one. However, the claim of altered coupling is based on 7 filopodia in 3D. I am afraid this is not adequate. The filopodia assay employed here allows in principle to measure hundreds of filopodia per cell. These figures should contain thousands of single filopodia data points (same is true for 3A and 2G). Especially so when there is such a large heterogeneity in filopodia response (3E). Also, the authors should state whether the data in 3D were selected/curated in any way.
9. The fitting of 3D assumes constant area/piezo and variable C0. Why? I would a priori expect that both change.
10. In 3E the “wash” appears to miss the “iso” as a simple negative control?
11. Line 260 reads “no change to sorting after hypo” however it refers to “% of filopodia that sort” and not to the “sorting coeff”, which are two very different metrics. Here I have to say I am amazed to see such an all/none response on per filopodia basis. Wonderful discovery and a great example of “macro-cooperativity”!

To end I want to stress that I really enjoyed reading this manuscript. It appears to be very well referenced, and various data are nicely “pulled in” to motivate and support the findings. The experimental approach appears to be a powerful combination of a high-throughput filopodia-curvature assay (ref 31) with single-membrane-tether-pulling (ref 18). This could have been made more clear in the introduction. Experiments and conditions are overall very well documented (on only few occasions the number of replicates or sample size is missing,

eg 2G).

Reviewer #2 (Remarks to the Author):

The paper "Membrane curvature governs the distribution of Piezo1 in cells" by Yang et al. describes work analyzing the distribution of the Piezo1 channels in cells, finding that the spatial distribution relates to membrane curvature. Inversely - given the channel changes its curvature upon activation (flattening from an intrinsically curved inactivated state structure to a more flat structure when fully activated) the distribution changes with the addition of an activator.

The paper treats a timely topic. It is well designed.

However, I have some major concerns, (a) with the representation of the literature and (b) with the curvature analysis.

Abstract, Citations in Abstract and throughout the paper:

While all important works/references are cited, the attribution of the works' meaning and significance is often entirely mistaken.

- Cryo-EM (refs 6-9) has not revealed that Piezo1 contains 38 helices (Line 24). To the best of my knowledge none of the current cryo-EM structures resolved 38 helices. Likely we know the number of helices from sequence analysis, structure prediction and from the Piezo2 structure.

- A flattened configuration was not evidenced by ref 10 (Line 26). Piezo flattening was first reported by HS-AFM (Lin, Nature 2019, ref 23). It has been confirmed by the more recent cryo-EM study in small liposomes (ref 10). The fact that ref 23 documented first the curved to flattened conformational change is also wrongly not mentioned in similar statements (Lines 265, 308). (Ref 23 is later cited in other inappropriate contexts)

- How the intrinsic curved architecture of Piezo makes the channel directly tension sensitive has not first been described by refs 11-13 (Line 28). It has first been rationalized in (Guo, eLife 2017; ref 7). (ref 7 is cited before with regard to structure only)

- How the large size and low density of helices impacts sensitivity is also better described in Guo, eLife 2017 and Lin, Nature 2019 rather than 11-13 (11,12 are not structural and therefore can barely provide primary data with regard to size).

- The curvature sensitivity and interactions with cytoskeletal components is not missing (Line 30). It has been described in (Dumitru Nano Letters 2021, ref 22). (Ref 22 is cited later, though it concerns cytoskeletal interactions)

- The authors write that the distribution of Piezo1 in cells are less well explored (than the structure and function, Lines 33 to 42), but reference eight detailed works (20 to 28, reference 23 should not be among them). I wonder if the paragraph setting up the question should not be rephrased? It is still an important topic, even if it has already been studied.

Experiments:

- Figure 1 and 2 and text: How do we know from figure 1A, 2AB that the Piezo channels have been trafficked well to the plasma membrane? The fluorescence signal is most intense where I expect the cell to be thickest, but the plasma membrane should be everywhere equal, one layer at the bottom of the cell, one on top. Some sort of confocal microscopy would likely be more conclusive.
- Most of the fluorescence images are highly over-saturated: mCherry channel 1A, right; GFP and mOrange in 1B. The saturation is capped as the normalized fluorescence bolts in 1C show.
- Due to the signal saturation and normalization I cannot evaluate the strength of the hPiezo1 signal. It seems extremely strong. Can the authors estimate a Piezo protein density? Should we assume that this is an expression 'effect' or is there evidence of a comparable density of Piezo1 in physiological settings?
- Figure 2D and text: Why is the low curvature tether so inhomogeneous? It looks like it has bubbles (also in the GPI channel). Is this always the case? Why?

The text part regarding Piezo curvature and membrane curvature:

Many things in this part of the paper remain unclear to me.

- Definitions: Since we talk about Piezo1 in a membrane, it is my understanding that we are talking about 2D-curvature and thus consider Gaussian curvature. In this case, a concave dip as a Piezo dome represents when viewed from the outside of the cell has also positive curvature (not negative, Line 182). Also Filopodia and Tethers would have one principle curvature being positive, and one being 0, and a Gaussian curvature of 0. The lax way that the authors work with curvature leads to me not understanding several aspects. Let me explain my problem: I am sure the authors agree that both filopodia and pulled tethers are only curved in one direction and have zero curvature along the long tube axis. In contrast, the piezo channel has curvature in both directions, it is a dome. However, the authors treat principle and Gaussian curvature as the same and mix the tube curvature values of filopodia/tethers and the dome curvature of Piezo in equation. I don't think that is correct, a dome has no interest to enter a tube, whatever the radius of the tube. Is there a strong argument that would allow the authors mix dome and tube curvature and draw quantitative conclusions.
- Line 181 and the following argument: No. The trimer of Piezo1 has only a 10nm radius dome in detergent or in a vesicle that has itself 10nm radius. It does not have 10nm radius of curvature in a flat non-tensed membrane, see Figure 3d, in Lin Nature, 2019, ref 23. This is an aspect that has not properly been adapted in ref 10. In this context, it seems like the authors' fit result (Line 199) gives a negative curvature of 5nm, and then they say that it compares well with the 10nm 'spherical invaginations' by cryo-EM. But the mentioned cryo-EM study also reports radius of curvature (not a 'spherical invaginations') and the radius is 10nm. Also, as mentioned above, the 10nm is not the relaxed radius of curvature of the protein-membrane system, which is likely ~17nm or slightly more.
- Line 229: The fact that the same analysis in the presence of Yoda fits to give an essentially

flat Piezo1 raises questions than it answers. This is in disagreement with molecular dynamics studies where Yoda binds to Piezo1, but does not lead to a flat structure (and likely failed attempts by all cryo-EM groups to get an open state structure by simply mixing Yoda to their protein).

Filopodia inhibition by Piezo:

The authors report that cells expression Piezo have less filopodia and essentially link it to a membrane physics (curvature disagreement) effect. Adding Yoda reverts the effect. While I follow the authors thinking, these experiments do not control for downstream signaling of Piezo, and the potential effect of Yoda, a membrane standing molecule, on the physical chemistry of the membrane. It appears that Piezo1 would have to be extremely dense to prohibit/induce morphological changes on the cell level. How does this correspond to copy numbers and copy per area estimates. The references to physiological extremes (eg 'knocking out Piezo in Drosophila promotes axon regeneration') could have any reason and the least likely is membrane physics. Could the authors check the hypothesis by pulling (or pushing) tethers from vesicles and cells with varying Piezo amounts?

Minor comments and typos:

- Title and elsewhere in the paper: The word "in cellulo" is wrong. The latin ablative following "in" of the feminine word is clearly "cellula".
- Line 173: Inset, not Insect
- In line 119, the authors mention that filopodia are actin-rich structures, while in line 129 the authors say filopodia have radii of 25 to 55 nm. How much actin can be in there? Are there any Super-resolution images that document actin in filopodia?
- in line 310: Abstain from using 'first'. Dumitru, Nano Letters 2021, came to quite similar conclusions before.

REVIEWER COMMENTS

Reviewer #1 (Remarks to the Author):

The MS of Yang et al. deals with depletion of human/mouse PIEZO1 from cellular protrusions (filopodia, and artificial tethers pulled directly from the plasma membrane/PM or from cell-attached Giant PM Vesicles/blebs). The depletion is always relative to membrane bound anchors (Caax, GPI) and is attributed to the preferential sorting of hPIEZO1 away from protrusions of high positive membrane curvature (+C). Yoda1 and hypo-osmotic pressure appear to regulate depletion suggesting the effect depends on the conformation of the mechanochannel. hPIEZO1 expression downregulates filopodia expression attributing curvature induction properties to PIEZO1.

This is a very interesting manuscript that is in principle relevant for NatComm because it aims to connect for the first time the remarkable deformations PIEZO1 induces to membranes, with tension (the classical functional regulator) and its sorting behaviour. However, in my opinion, unfortunately, there are a number of major/critical issues that have to be addressed before the main claims are validated and thus before considering publication.

We thank the reviewer for the accurate summary of our manuscript. We are glad that the reviewer recognizes the novelty and significance of our study. We address each of the concerns below (reviewer comments in black, response in blue, quoted changes in the manuscript highlighted in yellow):

#1.1. PM Distribution \neq organelle specific depletion. This MS deals exclusively with PIEZO1 1) depletion in 2) filopodia/protrusions. However, these two “corner stones of the findings” are entirely absent from the title, the abstract and the discussion. The sentence “regulator of distribution to the PM of living cells” does not appear to be justified by the data which show sorting from/to a specific “organelle” i.e. filopodia. The term “distribution”, which typically refers to “upconcentration”, appears also inappropriate for data that only show “varying degrees of depletion”. In light of the above, the authors may wish to consider focusing/emphasising in the discussion the potential biological role of PIEZO1 depletion from filopodia.

We agree with the reviewer that in our previous manuscript, the sorting of Piezo1 was only demonstrated as various levels of depletion from membrane protrusions. The prediction of Piezo1 sorting on membrane invaginations was mainly based on extrapolations of a parsimonious model (current Fig. S10) that fits the protrusion data.

Now, we take this opportunity to verify the key prediction from the extrapolated model - that Piezo1 would enrich towards ~ 100 nm radius cell membrane invaginations. To achieve this, we utilized a recent development in nanotechnology, pioneered by Wenting Zhao and Bianxiao Cui's labs (Lou et al., 2019; Zhao et al., 2017). An illustration of the experimental design and detailed findings are summarized in the current Fig. 3 and briefly discussed below.

In collaboration with Wenting Zhao's lab, we cultured cells on precisely engineered nanobars with curved ends and flat central regions. For a labelled membrane protein of interest, the end-to-center fluorescence ratio would report the protein's curvature sorting ability. We find that Piezo1 enriches to the curved ends of nanobars, whereas membrane marker signals are homogeneous across the entire nanobar (Fig. 3). The finding achieved strong statistical significance via hundreds of repeats on nanobars of the exact same geometry, a major technical strength of our chosen system. Furthermore, the enrichment of Piezo1 was observed on nanobars with 3 different curvatures (corresponding to diffraction-limited radii between 100 to 200 nm) and qualitatively agrees with our model (current Fig. S10). While further investigations on a wider range of membrane curvature are required to fully map out the sorting of Piezo1 on membrane invaginations, our data in the current Fig. 3 clearly verifies the prediction that membrane curvature can lead to enrichment of Piezo1 on cellular invaginations.

We now refer to this new finding in the Abstract, along with the previously observed depletion of Piezo1 on filopodia. We present a detailed description of the experiment and associated findings in the Results and the Method sections. We believe these new experiments and discussions, along with our original findings, are sufficient to support our claim that “*membrane curvature governs the distribution of Piezo1 in live cells*”.

#1.2.1 Fig. 1 aims to demonstrate that average PIEZO1 filopodia densities are smaller than GPI, CaaX, D2R, TREK-1. However, if protein overexpression affects filopodia size distribution this comparison is meaningless (since densities are hypothesised to be curvature depended). This does not appear to be a problem for PIEZO1 but seems to be a problem e.g. for D2R. How about TREK? Please add in 1E filopodia size distributions for all used constructs.

We thank the reviewer for pointing out this potential confusion. We have now quantified the radius of TREK1 expressing cells and the results are summarized in the updated Fig. 1E and Fig. S6C. TREK1 behaves similarly to D2R in that they enrich to filopodia ($S_{\text{filo}} > 1$) and that the radii of filopodia from TREK1/D2R expressing cells are smaller than the control conditions.

We note the reduction in filopodia radii is consistent with the observation that many curvature sensing proteins can also generate membrane curvature (i.e., impose the curvature of the protein to the membrane) (Baumgart et al., 2011). The ability to generate membrane curvature often requires a sufficient density of the curvature sensing protein, as has been quantitatively demonstrated for amphiphysin (Sorre et al., 2012) and for endophilin (Shi and Baumgart, 2015).

However, we do agree with the reviewer that in the scope of our current manuscript, changes in filopodia radii complicates the interpretation of D2R and TREK1's curvature sorting measurements. We now acknowledge this limitation in the main text. The current manuscript reads: “cells overexpressing D2R or TREK1 showed significantly reduced filopodia radii, consistent with the membrane deformation and curvature generation ability of many curvature sensing proteins (Baumgart et al., 2011; Shi and Baumgart, 2015). However, the molecular mechanisms and the causalities between the increased S_{filo} and the reduced filopodia radii in D2R or TREK1 expressing cells remain to be explored.”

#1.2.2. PS, I would have shown a zoom of the same area from 1A in the two channels to highlight there is no PIEZO1 signal.

We now show the zoom-in images of Fig. 1A and 1B as separate channels in the updated Fig. 1A, 1B, and in Fig. S1.

#1.3. Fig. 2 aims to establish that depletion scales with tether radius. 2E shows that 1) bleb-tethers from floppy (low tension) blebs have 2) larger diameters and 3) smaller depletion than other bleb-tethers (cell-tethers and filopodia). It is not clear to me why/how the authors deconvolve the possible influence of tension from curvature and conclude that only the latter is important (see also line 316).

We thank the reviewer for pointing out this confusion in our manuscript. We now explain the deconvolution between tension and curvature effects in detail. We also performed additional experiments to quantify the membrane tension in cells and blebs (current Fig. S9).

In the Results section, we add: “Tethers are typically imaged > 1 min after pulling, whereas membrane tension equilibrates within 1 s across cellular scale free membranes (e.g., bleb, tether) (Shi et al., 2018). Therefore, the sorting of Piezo1 within individual tension-equilibrated tether-bleb systems (Fig. 2C – 2G) suggests that membrane curvature can directly modulate Piezo1 distribution beyond potential confounding tension effects.”

In the Discussion section, we add: “In addition to membrane curvature, tension in the membrane may affect the subcellular distribution of Piezo1 (Dumitru et al., 2021). Particularly, membrane tension can activate the channel and potentially change Piezo1's nano-geometries. This tension effect is unlikely to play a significant role in our interpretation of the curvature sorting

of Piezo1 (Fig .2): (1) HeLa cell membrane tension as probed by short tethers (Fig. S9F; 45 ± 29 pN/ μm on blebs and 270 ± 29 pN/ μm on cells, with the highest recorded tension at 426 pN/ μm) are significantly lower than the activation tension for Piezo1 (> 1000 pN/ μm (Cox et al., 2016; Lewis and Grandl, 2015; Shi et al., 2018; Syeda et al., 2016)). (2) With more activated (and potentially flatten) channels under high membrane tension, one would expect a higher density of Piezo1 on tethers pulled from tensor blebs. This is the opposite to our observations in Fig. 2C - 2G, where Piezo1 density on tethers was found to decrease with the absolute curvature, thus tension (eq. S6), of membrane tethers.”

Regarding the model:

#1.4.1. This is very simple phenomenological model. It is perfectly fine to use phenomenological models to fit experimental data, because obtaining a good fit (for reasonable parameters) validates the model. However, in my opinion, such models should not be used to extrapolate to areas where no data have ever been collected (negative curvature in this case, S8) and where therefore no validation can be obtained.

We agree with the reviewer that the extrapolation of the model was not directly validated in our previous manuscript. To address this concern, we have now validated the key predictions from the model regarding the sorting of Piezo1 on membrane invaginations, using a recently developed nanobar technology. Please refer to our response to Comment #1.1 and the current Fig. 3 for details about the new data.

Additionally, we now plot the membrane invagination part of the model in dashed lines to indicate that it is extrapolated from the measurements on protrusions (in current Fig. S10). Please also note that per advice from reviewer #2, we now refer to invaginations as positive membrane curvature and protrusions as negative membrane curvature.

#1.4.2. Especially for such complex giga-Dalton proteins as the PIEZO, which has furthermore an inherently asymmetric structure and thus likely asymmetric behavior as a function of curvature.

The reviewer is correct that there is no reason to assume Piezo1 (or technically the Piezo1-membrane complex, see response to #1.5) would present the same spontaneous curvature on positively and negatively curved membranes. We now acknowledge this possibility in the Discussion section: “However, it is worth noting that we assumed a zero spontaneous curvature for membranes associated with Piezo1 and that the spontaneous curvature of Piezo1-membrane complex is independent of the shape of surrounding membranes. These assumptions may no longer hold when studying Piezo1 in highly curved invaginations or liposomes (Lin et al., 2019).”

#1.5. There appears to be a crucial misunderstanding. The spontaneous radius of curvature that comes out of the model is the curvature at which the membrane-protein-mismatch potential energy would be minimized. This is equal to the actual “nano-geometry in situ” of protein at the PM only! if this protein dominates over all competing membrane deformation/stabilisation processes. This however cannot be assumed, it rather has to be proved. In my opinion/understanding the data presented do not resolve nor provide any info regarding the actual membrane geometries around PIEZO1.

We agree with the reviewer that spontaneous curvature from our model C_0 ($C_0^{-1} = 83 \pm 17$ nm, the value is updated after refitting to more data points collected for Fig. 2G) represents a balance between the intrinsic curvature of Piezo1 trimers ($0.04 \sim 0.2$ nm $^{-1}$ as suggested by CryoEM studies (Haselwandter, MacKinnon et al., 2022; Lin et al., 2019; Yang et al., 2022)) and that of the associated membrane (0 nm $^{-1}$, assuming lipid bilayers alone do not have an intrinsic curvature).

We now refer to C_0 as the “spontaneous curvature of the Piezo1-membrane complex” throughout the manuscript, rather than the “spontaneous curvature of Piezo1”. As the majority of plasma membrane is flat, we believe that the C_0 value from our model is biophysically relevant for evaluating the behavior of Piezo1 in cells. We now clarify the physical meaning of C_0 when

presenting the model (eq. 1 and Method). We also discuss limitations in our simplistic model and their effects on the interpretation of C_0 .

The relative bending stiffness between Piezo1 and the lipid membrane indeed play a role in our model by modulating the bending stiffness of the protein-membrane complex ($\tilde{\kappa}$). We realized that this aspect was incorrectly presented in our previous manuscript, after discussion with experts (Elizabeth Kelley and Markus Deserno) at a membrane mechanics conference. The corrected relation is shown as the updated eq. S11 in the revised manuscript. This correction does not change the interpretation of either the curvature sorting model or our data. On a related note, Piezo1 was recently found to be (surprisingly) flexible, with a bending stiffness similar to that of the lipid membrane (Haselwandter, Guo et al., 2022).

#1.6. If I understood correctly the model provided a spontaneous radius of curvature of -87 nm. This appears to be an order of magnitude larger than the ~10nm obtained by CryoEM, however the author suggest the numbers are in good agreement (line 205). Sorry if I misunderstood something...?

As noted in the response to comment #1.5., $C_0^{-1} = 83 \pm 17$ nm represents a balance between the intrinsic curvature of Piezo1 trimers ($0.04 \sim 0.2$ nm⁻¹ as suggested by CryoEM studies(Haselwandter et al., 2022; Lin et al., 2019; Yang et al., 2022)) and that of the associated membrane (0 nm⁻¹, assuming lipid bilayers alone do not have an intrinsic curvature). CryoEM images Piezo1 in highly curved liposomes that can confine the shape of the protein, the associated apparent curvature of Piezo1 therefore represents an upper bound for C_0 in our model.

Of note, MacKinnon's lab recently suggested that Piezo1 would take on a radius of curvature ~ 42 nm when reconstituted into purified flat lipid bilayers (Haselwandter et al., 2022; Haselwandter et al., 2022), significantly lower than the apparent curvature of Piezo1 in ~ 10 nm liposomes.

We now clarify these points when presenting the fitting results from our model: " C_0 , expected to be positive for Piezo1, is the spontaneous curvature of each protein-membrane complex in the (mostly flat) plasma membrane and would be lower than the apparent curvature of Piezo1 in detergents or in highly curved liposomes (Haselwandter et al., 2022; Lin et al., 2019; Yang et al., 2022)."

#1.7. The model is using bending stiffness of cellular membranes however, it is fitting bleb data. Isn't more appropriate to measure directly the bending stiffness of blebs?

We have now measured the bending stiffness of bleb membranes (12.7 ± 2.5 k_BT, $n = 14$; Fig. S9) and updated the corresponding estimate to the area of each Piezo1 unit ($A_P = 380 \pm 170$ nm²). The bending stiffness of blebs is similar to that of the cell membrane (13.6 ± 3.8 k_BT), consistent with it being an intrinsic material property that mainly depends on lipid compositions (Deserno, 2007).

Figure 3

#1.8.1. Fig. 3 aims to demonstrate that yoda1 application which activates the channel, changes its coupling to curvature. This is a crucial figure in the manuscript, in my opinion the most important one. However, the claim of altered coupling is based on 7 filopodia in 3D. I am afraid this is not adequate. The filopodia assay employed here allows in principle to measure hundreds of filopodia per cell. These figures should contain thousands of single filopodia data points (same is true for 3A and 2G). Especially so when there is such a large heterogeneity in filopodia response (3E).

We thank the reviewer for recognizing the importance of our findings. We have now increased the statistics in the current Fig. 4B -4D, to 66 filopodia. We updated the p values in Fig. 4B, 4C and updated the fitting results in Fig. 4D. The relevant conclusions remain unchanged.

We also would like to point out that only a fraction of filopodia can be feasibly used for pairwise comparisons, as filopodia often change position during the 10 min waiting period after addition of

Yoda1. Additionally, only filopodia that are sufficiently long and separated from each other are used for sorting and radius quantifications. These aspects are now detailed in the Method section and in the current Fig. S3. Therefore, although it is relatively easy to image thousands of filopodia, we currently can only achieve accurate quantifications on a small fraction of those imaged filopodia.

We have also increased the number of membrane bleb data points in Fig. 2G from 24 to 38. We would like to highlight the fact that these data were obtained from manually pulled tethers and often take days to weeks for an expert to prepare and measure a handful of such data points.

Lastly, the newly added data regarding enrichment of Piezo1 on membrane invaginations (Fig. 3) was collected from ~400 individual nanobar-induced membrane invaginations.

#1.8.2. Also, the authors should state whether the data in 3D were selected/curated in any way.

The current Fig. 4D only contains data under the regular condition (without hypotonic shock, with Ca^{2+} , noted in the current Fig. 4E as Iso+ Ca^{2+}), as hypotonic shocks often induce a slight swelling of the cell body, making accurate filopodia radius measurements challenging. We now note this in the caption of current Fig. 4: “All data points in (B) - (D) are quantified from cells in regular XC and are limited to filopodia that did not significantly change positions after addition of Yoda1”

#1.9. The fitting of 3D assumes constant area/piezo and variable C0. Why? I would a priori expect that both change.

In our model, A_p represents the area of the curved surface for each Piezo1-membrane complex. Since the sequence and the trimeric structure of the protein are not expected to change with addition of Yoda1, we kept A_p as a constant. Additionally, having 1 free parameter reduces the uncertainty in fitting, this is particularly important as the range of filopodium radius in the current Fig. 4D is relatively small, an aspect that would not benefit from measuring more filopodia.

We now clarify this after eq. (1): “ A_p represents the area of a potentially curved surface and should not be confused with the projected area of the protein.”

#1.10. In 3E the “wash” appears to miss the “iso” as a simple negative control?

We thank the reviewer for this suggestion. We have now performed the suggested negative control by flowing isotonic buffer to Piezo1 expressing cells (current Fig. S12D).

The text in the Results section was updated to: “Additionally, Yoda1 induced sorting of Piezo1 is reversible as Piezo1 signals disappear from filopodia after washing out Yoda1 (Fig. 4E-G). Importantly, shear stress applied during washing steps alone did not significantly change the sorting of Piezo1 on filopodia (Fig. S12D).”

#1.11.1. Line 260 reads “no change to sorting after hypo” however it refers to “% of filopodia that sort” and not to the “sorting coeff”, which are two very different metrics.

We thank the reviewer for pointing this out. We have updated the main text to “The hypotonic shock itself did not significantly change the fraction of filopodia that showed significant sorting of Piezo1.”

#1.11.2. Here I have to say I am amazed to see such an all/none response on per filopodia basis. Wonderful discovery and a great example of “macro-cooperativity”!

This is a great point! We now mention this after presenting Fig. 4: “On individual filopodium, the apparent all-or-none response to Yoda1 treatment suggests that cooperativity between Yoda1-Piezo1 or Piezo1-Piezo1 may also play a role in modulating the channel’s curvature sorting behavior.”

#1.11.1. To end I want to stress that I really enjoyed reading this manuscript. It appears to be very

well referenced, and various data are nicely “pulled in” to motivate and support the findings. The experimental approach appears to be a powerful combination of a high-throughput filopodia-curvature assay (ref 31) with single-membrane-tether-pulling (ref 18). This could have been made more clear in the introduction.

We thank the reviewer for the positive assessment of our manuscript and for the kind suggestion. The update introduction reads: “Here, we combine high-throughput filopodia (Rosholm et al., 2017) and nanobar (Lou et al., 2019; Zhao et al., 2017) sorting assays with quantitative single membrane tether pulling experiments (Shi et al., 2018) to show experimentally that membrane curvature is a fundamental regulator of Piezo1’s distribution within the plasma membrane.”.

#1.11.2. Experiments and conditions are overall very well documented (on only few occasions the number of replicates or sample size is missing, eg 2G).

We added statistics on sample size in Fig. 2G and double checked the label of sample size in other figure panels.

Reviewer #2 (Remarks to the Author):

The paper "Membrane curvature governs the distribution of Piezo1 in cellulo" by Yang et al. describes work analyzing the distribution of the Piezo1 channels in cells, finding that the spatial distribution relates to membrane curvature. Inversely - given the channel changes its curvature upon activation (flattening from an intrinsically curved inactivated state structure to a more flat structure when fully activated) the distribution changes with the addition of an activator.

The paper treats a timely topic. It is well designed. However, I have some major concerns, (a) with the representation of the literature and (b) with the curvature analysis.

We thank the reviewer for the accurate summary and overall positive assessment of our study. We address each of the concerns below (reviewer comments in black, response in blue, quoted changes in the manuscript highlighted in yellow):

Abstract, Citations in Abstract and throughout the paper:

While all important works/references are cited, the attribution of the works' meaning and significance is often entirely mistaken.

#2.1. Cryo-EM (refs 6-9) has not revealed that Piezo1 contains 38 helices (Line 24). To the best of my knowledge none of the current cryo-EM structures resolved 38 helices. Likely we know the number of helices from sequence analysis, structure prediction and from the Piezo2 structure.

We thank the reviewer for pointing out this inaccuracy in our manuscript. The revised introduction now reads: “Each Piezo1 monomer contains 38 transmembrane helices and cryogenic electron microscopy (CryoEM) shows a trimeric propeller-like assembly which is thought to curve into the cytosol in its resting state”

#2.2. A flattened configuration was not evidenced by ref 10 (Line 26). Piezo flattening was first reported by HS-AFM (Lin, Nature 2019, ref 23). It has been confirmed by the more recent cryo-EM study in small liposomes (ref 10). The fact that ref 23 documented first the curved to flattened conformational change is also wrongly not mentioned in similar statements (Lines 265, 308). (Ref 23 is later cited in other inappropriate contexts)

We thank the reviewer for pointing out the missed referencing of Lin et al. We now cite Lin et al while referencing the potential flattening of Piezo1 and several other relevant places throughout the text.

The first appearance of the citation reads: “Recently, a flattened configuration of Piezo1 was identified when reconstituted into small liposomes, potentially corresponding to the open/inactivated state of the ion channel (Yang et al., 2022) and confirming previous work using high speed atomic force microscopy (HS-AFM) (Lin et al., 2019).”

#2.3. How the intrinsic curved architecture of Piezo makes the channel directly tension sensitive has not first been described by refs 11-13 (Line 28). It has first been rationalized in (Guo, eLife 2017; ref 7). (ref 7 is cited before with regard to structure only)

We now cite Guo et al while referencing to how the intrinsically curved architecture of Piezo promotes the channel’s tension sensitive.

#2.4. How the large size and low density of helices impacts sensitivity is also better described in Guo, eLife 2017 and Lin, Nature 2019 rather than 11-13 (11,12 are not structural and therefor can barely provide primary data with regard to size).

We cite refs 11 and 12 as they provide direct evidence to support that Piezo1 is sensitive to membrane tension. We agree with the reviewer that Guo et al and Lin et al better describes the effect of protein size on the gating of Piezo1.

#2.5. The curvature sensitivity and interactions with cytoskeletal components is not missing (Line 30). It has been described in (Dumitru Nano Letters 2021, ref 22). (Ref 22 is cited later, though it concerns cytoskeletal interactions)

We changed the introduction to: “several studies indicate a cytoskeletal role in the activation of Piezo channels, while the presence of direct Piezo1-cytoskeleton interaction is still under debate (Bavi et al., 2019; Dumitru et al., 2021; Mylvaganam et al., 2022; Romero et al., 2020; Vaisey et al., 2022; Verkest et al., 2022; Wang et al., 2022).”

We now cite Dumitru et al as ref 21 but should point out to the reviewer that the Dumitru et al article does not provide robust evidence that Piezo1 is bound to the cytoskeleton. Consistent with this, we highlight a recent study from the MacKinnon lab (Vaisey et al, 2022), where no Piezo1-cytoskeleton interaction was observed in red blood cells.

We now also cite Dumitru et al when introducing the study of Piezo distribution in cells.

#2.6. The authors write that the distribution of Piezo1 in cells are less well explored (than the structure and function, Lines 33 to 42), but reference eight detailed works (20 to 28, reference 23 should not be among them). I wonder if the paragraph setting up the question should not be rephrased? It is still an important topic, even if it has already been studied.

We thank the reviewer for recognizing the importance of our study. We note that interesting subcellular distribution of Piezo1 has been observed in several studies, but a general mechanistic explanation has been missing. We rephrased the introduction to: “the dynamics and distribution of the channel within a cell are only starting to be explored (Dumitru et al., 2021; Ellefsen et al., 2019; Vaisey et al., 2022; Yao et al., 2020).” and “While a general mechanistic explanation has yet to be established, these intriguing subcellular patterns of Piezo1 raise the question of whether this ion channel can be sorted by fundamental physical factors on the cell surface.”

Experiments:

#2.7. - Figure 1 and 2 and text: How do we know from figure 1A, 2AB that the Piezo channels have been trafficked well to the plasma membrane? The fluorescence signal is most intense

where I expect the cell to be thickest, but the plasma membrane should be everywhere equal, one layer at the bottom of the cell, one on top. Some sort of confocal microscopy would likely be more conclusive.

We now have several lines of evidence to show that Piezo1 has been well trafficked to the plasma membrane.

1. We did additional confocal (current Fig. S1B) and TIRF (current Fig. S1C) imaging to show that Piezo1-eGFP signals are localized to the boundary of the cell as well as to the basal membrane.
2. On the cell body, the intensity profile of Piezo1-eGFP is similar to that of GPI-mCherry. GPI is a well-documented plasma membrane marker (Fig. 1A, 2A, S1A). Fluorescence intensity profile of Piezo1 follows that of the membrane markers on a flat region of the cell body (Fig. 1C). As the reviewer pointed out, the intensity is essentially constant on flat membranes.
3. Piezo1 is clearly present on the boundary of membrane blebs, which are plasma membranes that are detached from the cytoskeleton. (Fig. 2C, 2D).

We have now updated the manuscript accordingly to make this point more conclusive. Of note, membrane proteins often have retention to intracellular organelles such as the ER, giving an apparent higher signal at the central region of the cells, especially when the protein is overexpressed.

#2.8. Most of the fluorescence images are highly over-saturated: mCherry channel 1A, right; GFP and mOrange in 1B. The saturation is capped as the normalized fluorescence bolts in 1C show.

The image contrast in Fig. 1A was adjusted to show the dim filopodia (in later figures, this was achieved by taking logarithm of the intensity). We now provide images under regular contrast in the current Fig. S1.

The intensity profiles in Fig. 1C do not contain any saturation. A zoom-in to the maxima of the normalized intensities are shown below. Please note that the intensities are shown in log-scale.

#2.9.1. Due to the signal saturation and normalization I cannot evaluate the strength of the hPiezo1 signal. It seems extremely strong.

Please see response to comment #2.8.

#2.9.2. Can the authors estimate a Piezo protein density? Should we assume that this is an expression 'effect' or is there evidence of a comparable density of Piezo1 in physiological settings?

In collaboration with the Pathak lab, we repeated our measurements of Piezo1 sorting with mouse embryonic fibroblasts (MEFs) where Piezo1 has been labelled endogenously with tdTomato. The results, which are indistinguishable from observations on Piezo1-overexpressed HeLa cells, are summarized in current Figs. 1D, 1E and S5. These new measurements confirm that the depletion of Piezo1 from membrane protrusions also happens at physiological condition.

Taking advantage of these MEFs, we also estimated that the amount of overexpressed Piezo1 in HeLa cells in Fig. 1 is about 2.5-fold of the endogenous level (Fig. S5C).

#2.10. Figure 2D and text: Why is the low curvature tether so inhomogeneous? It looks like it has bubbles (also in the GPI channel). Is this always the case? Why?

We thank the reviewer for pointing out this interesting observation. While detailed molecular mechanisms of the inhomogeneity are still unclear, we hypothesize that this is due to potential clustering of Piezo1. We now discuss this aspect: “Notably, apparent clusters of Piezo1 were often observed on cell membranes (Fig. 1, Fig. S1- S5) and on membrane tethers (Fig. 2D), suggesting that attractions between Piezo1 trimers may also be important for controlling the detailed distribution of the channel in cells.”

The text part regarding Piezo curvature and membrane curvature:
Many things in this part of the paper remain unclear to me.

#2.11. Definitions: Since we talk about Piezo1 in a membrane, it is my understanding that we are talking about 2D-curvature and thus consider Gaussian curvature. In this case, a concave dip as a Piezo dome represents when viewed from the outside of the cell has also positive curvature (not negative, Line 182).

We have now changed the definition of positive and negative curvature to follow the convention in the literature. Please note that surfaces that curve towards opposite directions (such as protrusions vs. invaginations on cell surface) have opposite curvatures.

#2.12. Also Filopodia and Tethers would have one principle curvature being positive, and one being 0, and a Gaussian curvature of 0. The lax way that the authors work with curvature leads to me not understanding several aspects.

As the reviewer noted, each 2D surface has two principal curvatures (C_1 and C_2). Membrane shape and its bending energy are often dominated by the mean curvature: $(C_1+C_2)/2$ (Deserno, 2007), because the Gaussian curvature (C_1*C_2) contributes to a constant that only depends on the boundary conditions (Gauss-Bonnet theorem). We have now clarified this aspect after presenting eq. S10.

#2.13. Let me explain my problem: I am sure the authors agree that both filopodia and pulled tethers are only curved in one direction and have zero curvature along the long tube axis. In contrast, the piezo channel has curvature in both directions, it is a dome. However, the authors treat principle and Gaussian curvature as the same and mix the tube curvature values of filopodia/tethers and the dome curvature of Piezo in equation. I don't think that is correct, a dome has no interest to enter a tube, whatever the radius of the tube. Is there a strong argument that would allow the authors mix dome and tube curvature and draw quantitative conclusions.

We thank the reviewer for bringing up this important point. By omitting Gaussian curvature in our model, we have also neglected potential effects of curvature anisotropy in regulating the sorting of Piezo1. However, including corrections from curvature anisotropy would greatly complicate our parsimonious model and is beyond the scope of the current investigation.

Additionally, note that mean curvature often dominates the sorting of curvature sensing proteins: proteins with isotropic curvature can sense the curvature of membrane tubes (Capraro et al., 2010), whereas proteins with anisotropic curvature (such as BAR domain proteins) can sense the curvature of spherical vesicles (Bhatia et al., 2009). In both cases, the experimental observations can be quantitatively explained by models that only consider the mean curvature of the membrane.

We now discuss this limitation: “Lastly, Piezo1-membrane complexes likely have isotropic curvatures (Guo and MacKinnon, 2017), whereas the membrane protrusions and invaginations in our study are anisotropically curved, with one principal membrane curvature close to zero. Improved models that consider the effect of curvature anisotropy may provide additional insights to the curvature sensing of Piezo1 (Alimohamadi et al., 2021; Kabaso et al., 2012).”

#2.14. Line 181 and the following argument: No. The trimer of Piezo1 has only a 10nm radius dome in detergent or in a vesicle that has itself 10nm radius. It does not have 10nm radius of curvature in a flat non-tensed membrane, see Figure 3d, in Lin Nature, 2019, ref 23. This is an aspect that has not properly been adapted in ref 10. In this context, it seems like the authors' fit result (Line 199) gives a negative curvature of 5nm, and then they say that it compares well with the 10nm 'spherical invaginations' by cryo-EM. But the mentioned cryo-EM study also reports radius of curvature (not a 'spherical invaginations') and the radius is 10nm. Also, as mentioned above, the 10nm is not the relaxed radius of curvature of the protein-membrane system, which is likely ~17nm or slightly more.

We thank the reviewer for bringing up this important point. Please see our response to comments #1.5. and #1.6.

#2.15. Line 229: The fact that the same analysis in the presence of Yoda fits to give an essentially flat Piezo1 raises questions than it answers. This is in disagreement with molecular dynamics studies where Yoda binds to Piezo1, but does not lead to a flat structure (and likely failed attempts by all cryo-EM groups to get an open state structure by simply mixing Yoda to their protein).

We thank the reviewer for providing helpful information regarding Yoda1. We think there are several possible reasons for the apparent discrepancy.

First of all, we note that only a small fraction of filopodia responded to Yoda1, and pre-stressing of the cell membrane is required to amplify the Yoda1 effect (current Fig. 4E). This observation suggests that membrane tension is likely required to flatten Piezo1, even in the presence of Yoda1. Secondly, as noted by the reviewer, highly curved liposome or detergents can confine the shape of Piezo1 trimers. Therefore, the inability to observe flattened Piezo1 *in vitro* is not necessarily in contradiction with our observation in the mostly flat cell membranes. To our knowledge, there has not been published results regarding Piezo1 structure in high-tension flat membranes such as stretched giant unilamellar vesicles. Lastly, potential flattening of Piezo1 by Yoda1 may take significantly longer than the runtime of current molecular dynamics simulations.

We add to the Discussion section: "Yoda1 induced flattening of Piezo1 has not been directly observed via CryoEM. Our results (Fig. 4) point to two challenges in determining this potential structural change: (1) Yoda1 induced changes in Piezo1 sorting is greatly amplified after pre-stretching the membrane (Fig. 4E), pointing to the possibility that a significant tension in the membrane is required for the flattening of Yoda1-bound Piezo1. (2) Piezo1 is often incorporated in small (< 20 nm radius) liposomes for CryoEM studies. The shape of liposomes can confine the nano-geometry of Piezo1 (Lin et al., 2019; Yang et al., 2022), rendering it significantly more challenging to respond to potential Yoda1 effects. This potential effect of membrane curvature on the activation of Piezo1 would be an interesting direction for future studies."

We also realize that our data do not directly show the flattening of Piezo1 by Yoda1, rather it is consistent with the flattening hypotheses. We lowered the tone of our conclusion to Fig. 4 to: "Our study suggests this conformational change of Piezo1 may also happen in live cells (Fig. 4H)." We also added arrows in Fig. 4H to suggest that membrane tension helps the proposed flattening of Piezo1 by Yoda1.

Filopodia inhibition by Piezo:

#2.16. The authors report that cells expression Piezo have less filopodia and essentially link it to a membrane physics (curvature disagreement) effect. Adding Yoda reverts the effect. While I follow the authors thinking, these experiments do not control for downstream signaling of Piezo, and the potential effect of Yoda, a membrane standing molecule, on the physical chemistry of the

membrane. It appears that Piezo1 would have to be extremely dense to prohibit/induce morphological changes on the cell level. How does this correspond to copy numbers and copy per area estimates. The references to physiological extremes (eg 'knocking out Piezo in Drosophila promotes axon regeneration') could have any reason and the least likely is membrane physics. Could the authors check the hypothesis by pulling (or pushing) tethers from vesicles and cells with varying Piezo amounts?

We thank the reviewer for bringing up this important point. We have performed extensive additional experiments to address the concern.

(1). We confirm that on Piezo1-KO cells, Yoda1 molecules alone do not significantly alter the formation of filopodia. The data is now presented as Fig. 5E in the updated manuscript.

(2). As suggested by the reviewer, we measured tether pulling force on hPiezo1 overexpressing HEK293T cells and compared it to HEK293T cells without the overexpression. hPiezo1 overexpression significantly increased the force needed to pull the tether, supporting a direct mechanical role exerted by Piezo1. These results are now presented as Fig. 5F – 5H.

(3). In collaboration with the Pathak lab, we compared the number of filopodia on MEFs dissected from wild type (WT) or Piezo1 heterozygous (Het.) mice with those from their Piezo1-KO littermates. Both WT and Het. MEFs showed significantly less filopodia compared to their Piezo1-KO counterparts, suggesting that endogenous Piezo1 can already inhibit filopodia formation. These results are now presented as Fig. 5I – 5K.

(4). As mentioned in the response to comment #2.9.2., we estimated that the amount of overexpressed Piezo1 in HeLa cells in Fig. 1 is about 2.5-fold of the endogenous level (Fig. S5C). By comparing Fig. 5B and 5D, we can further estimate that the overexpression of Piezo1 in HEK293T cells are ~ 10-fold that of the endogenous level.

Additionally, we would like to draw the reviewer's attention to a recent publication by the Luo lab. Where it was observed that in Drosophila, Piezo can modulate the morphogenesis and targeting of dendrites independently of its mechanosensitive ion channel activity (Xie et al., 2022). Finally, we would like to point out that our data do not exclude other mechanisms that control filopodia formation, such as Piezo1 induced Ca^{2+} signaling. Rather, we suggest that the curvature preference of Piezo1 provides an additional route to control filopodia dynamics, in parallel to other potential mechanisms.

We now discuss these points in our updated manuscript. We also revised our conclusion: "In addition to membrane curvature, several parallel Piezo-related mechanisms can regulate the formation and growth of filopodia, including Ca^{2+} signaling induced by the activation of Piezo and potential interactions between specific Piezo domains with cytoskeletal components (Ma et al., 2022; Song et al., 2019; Verkest et al., 2022). Our data do not exclude these parallel mechanisms, rather, we suggest that the curvature preference of Piezo1 provides an additional route to control filopodia dynamics. Further studies are required to fully dissect the contribution of each of these variables."

Minor comments and typos:

#2.17. Title and elsewhere in the paper: The word "in cellulo" is wrong. The latin ablative following "in" of the feminine word is clearly "cellula".

We thank the reviewer for pointing out our poor use of Latin. We have changed '*in cellulo*' to 'in live cells'.

#2.18. Line 173: Inset, not Insect.
Corrected.

#2.19. In line 119, the authors mention that filopodia are actin-rich structures, while in line 129 the authors say filopodia have radii of 25 to 55 nm. How much actin can be in there? Are there any Super-resolution images that document actin in filopodia?

Filopodia typically have “10-30 actin filaments held together by actin-binding proteins” according to MBINFO. Super-resolution images of actin in filopodia can be found, for example, in Fig. 2 of the following paper:

Sudhaharan, Thankiah, et al. "Superresolution microscopy reveals distinct localisation of full length IRSp53 and its I-BAR domain protein within filopodia." Scientific reports 9.1 (2019): 1-17.

#2.20. in line 310: Abstain from using 'first'. Dumitru, Nano Letters 2021, came to quite similar conclusions before.

Removed 'first'.

Reference List

Alimohamadi, H., Bell, M.K., Halpain, S., and Rangamani, P. (2021). Mechanical principles governing the shapes of dendritic spines. *Frontiers in Physiology* 836.

Baumgart, T., Capraro, B.R., Zhu, C., and Das, S.L. (2011). Thermodynamics and mechanics of membrane curvature generation and sensing by proteins and lipids. *Annu. Rev. Phys. Chem.* 62, 483.

Bavi, N., Richardson, J., Heu, C., Martinac, B., and Poole, K. (2019). PIEZO1-mediated currents are modulated by substrate mechanics. *ACS Nano* 13, 13545-13559.

Bhatia, V.K., Madsen, K.L., Bolinger, P., Kunding, A., Hedegård, P., Gether, U., and Stamou, D. (2009). Amphipathic motifs in BAR domains are essential for membrane curvature sensing. *Embo J.* 28, 3303-3314.

Capraro, B.R., Yoon, Y., Cho, W., and Baumgart, T. (2010). Curvature sensing by the epsin N-terminal homology domain measured on cylindrical lipid membrane tethers. *J. Am. Chem. Soc.* 132, 1200-1201.

Cox, C.D., Bae, C., Ziegler, L., Hartley, S., Nikolova-Krstevski, V., Rohde, P.R., Ng, C., Sachs, F., Gottlieb, P.A., and Martinac, B. (2016). Removal of the mechanoprotective influence of the cytoskeleton reveals PIEZO1 is gated by bilayer tension. *Nature Communications* 7, 1-13.

Deserno, M. (2007). Fluid lipid membranes—a primer. See [Http://www.Cmu.Edu/Biolphys/Deserno/Pdf/Membrane_theory.Pdf](http://www.Cmu.Edu/Biolphys/Deserno/Pdf/Membrane_theory.Pdf)

Dumitru, A.C., Stommen, A., Koehler, M., Cloos, A., Yang, J., Leclercqz, A., Tyteca, D., and Alsteens, D. (2021). Probing PIEZO1 Localization upon Activation Using High-Resolution Atomic Force and Confocal Microscopy. *Nano Letters* 21, 4950-4958.

Ellefsen, K.L., Holt, J.R., Chang, A.C., Nourse, J.L., Arulmoli, J., Mekhdjian, A.H., Abuwarda, H., Tombola, F., Flanagan, L.A., Dunn, A.R., Parker, I., and Pathak, M.M. (2019). Myosin-II mediated traction forces evoke localized Piezo1-dependent Ca²⁺ flickers. *Communications Biology* 2, 1-13.

Guo, Y.R., and MacKinnon, R. (2017). Structure-based membrane dome mechanism for Piezo mechanosensitivity. *Elife* 6, e33660.

Haselwandter, C.A., Guo, Y., Fu, Z., and MacKinnon, R. (2022). Elastic properties and shape of the Piezo dome underlying its mechanosensory function. *bioRxiv*

Haselwandter, C.A., MacKinnon, R., Guo, Y., and Fu, Z. (2022). Quantitative prediction and measurement of Piezo's membrane footprint. *bioRxiv*

Kabaso, D., Bobrovska, N., Gózdź, W., Gov, N., Kralj-Iglič, V., Veranič, P., and Iglič, A. (2012). On the role of membrane anisotropy and BAR proteins in the stability of tubular membrane structures. *J. Biomech.* 45, 231-238.

Lewis, A.H., and Grandl, J. (2015). Mechanical sensitivity of Piezo1 ion channels can be tuned by cellular membrane tension. *Elife* 4, e12088.

Lin, Y., Guo, Y.R., Miyagi, A., Levring, J., MacKinnon, R., and Scheuring, S. (2019). Force-induced conformational changes in PIEZO1. *Nature* 573, 230-234.

Lou, H., Zhao, W., Li, X., Duan, L., Powers, A., Akamatsu, M., Santoro, F., McGuire, A.F., Cui, Y., and Drubin, D.G. (2019). Membrane curvature underlies actin reorganization in response to nanoscale surface topography. *Proceedings of the National Academy of Sciences* 116, 23143-23151.

Ma, N., Chen, D., Lee, J., Kuri, P., Hernandez, E.B., Kocan, J., Mahmood, H., Tichy, E.D., Rompolas, P., and Mourkioti, F. (2022). Piezo1 regulates the regenerative capacity of skeletal muscles via orchestration of stem cell morphological states. *Science Advances* 8, eabn0485.

Mylvaganam, S., Plumb, J., Yusuf, B., Li, R., Lu, C., Robinson, L.A., Freeman, S.A., and Grinstein, S. (2022). The spectrin cytoskeleton integrates endothelial mechanoresponses. *Nat. Cell Biol.* 24, 1226-1238.

Romero, L.O., Caires, R., Nickolls, A.R., Chesler, A.T., Cordero-Morales, J.F., and Vásquez, V. (2020). A dietary fatty acid counteracts neuronal mechanical sensitization. *Nature Communications* 11, 1-12.

Rosholm, K.R., Leijnse, N., Mantsiou, A., Tkach, V., Pedersen, S.L., Wirth, V.F., Oddershede, L.B., Jensen, K.J., Martinez, K.L., Hatzakis, N.S., *et al.* (2017). Membrane curvature regulates ligand-specific membrane sorting of GPCRs in living cells. *Nature Chemical Biology* 13, 724-729.

Shi, Z., and Baumgart, T. (2015). Membrane tension and peripheral protein density mediate membrane shape transitions. *Nature Communications* 6, 1-8.

Shi, Z., Graber, Z.T., Baumgart, T., Stone, H.A., and Cohen, A.E. (2018). Cell membranes resist flow. *Cell* 175, 1769-1779. e13.

- Song, Y., Li, D., Farrelly, O., Miles, L., Li, F., Kim, S.E., Lo, T.Y., Wang, F., Li, T., and Thompson-Peer, K.L. (2019). The mechanosensitive ion channel piezo inhibits axon regeneration. *Neuron* 102, 373-389. e6.
- Sorre, B., Callan-Jones, A., Manzi, J., Goud, B., Prost, J., Bassereau, P., and Roux, A. (2012). Nature of curvature coupling of amphiphysin with membranes depends on its bound density. *Proceedings of the National Academy of Sciences* 109, 173-178.
- Syeda, R., Florendo, M.N., Cox, C.D., Kefauver, J.M., Santos, J.S., Martinac, B., and Patapoutian, A. (2016). Piezo1 channels are inherently mechanosensitive. *Cell Reports* 17, 1739-1746.
- Vaisey, G., Banerjee, P., North, A.J., Haselwandter, C.A., and MacKinnon, R. (2022). Piezo1 as a force-through-membrane sensor in red blood cells. *bioRxiv*
- Verkest, C., Schaefer, I., Nees, T.A., Wang, N., Jegelka, J.M., Taberner, F.J., and Lechner, S.G. (2022). Intrinsically disordered intracellular domains control key features of the mechanically-gated ion channel PIEZO2. *Nature Communications* 13, 1-14.
- Wang, J., Jiang, J., Yang, X., Zhou, G., Wang, L., and Xiao, B. (2022). Tethering Piezo channels to the actin cytoskeleton for mechanogating via the cadherin- β -catenin mechanotransduction complex. *Cell Reports* 38, 110342.
- Xie, Q., Li, J., Li, H., Udeshi, N.D., Svinkina, T., Orlin, D., Kohani, S., Guajardo, R., Mani, D.R., Xu, C., *et al.* (2022). Transcription factor Acj6 controls dendrite targeting via a combinatorial cell-surface code. *Neuron*
- Yang, X., Lin, C., Chen, X., Li, S., Li, X., and Xiao, B. (2022). Structure deformation and curvature sensing of PIEZO1 in lipid membranes. *Nature* 1-7.
- Yao, M., Tijore, A.S., Cox, C.D., Hariharan, A., Van Nhieu, G.T., Martinac, B., and Sheetz, M. (2020). Force-dependent Piezo1 recruitment to focal adhesions regulates adhesion maturation and turnover specifically in non-transformed cells. *bioRxiv*
- Zhao, W., Hanson, L., Lou, H., Akamatsu, M., Chowdary, P.D., Santoro, F., Marks, J.R., Grassart, A., Drubin, D.G., and Cui, Y. (2017). Nanoscale manipulation of membrane curvature for probing endocytosis in live cells. *Nature Nanotechnology* 12, 750-756.

REVIEWERS' COMMENTS

Reviewer #2 (Remarks to the Author):

The authors have provided detailed responses to my comments. I appreciate the addition of additional experiments. Considering the simplifications of the physical model when treating curvature and potential misunderstandings regarding the membrane-protein curvature (my comments 2.11 to 2.15 and referee #1 comments 1.5 to 1.7) and the various measurement uncertainties, I would abstain from strong quantitative statements, and rather keep the message qualitative as state-dependent protein sorting according to membrane geometries.

REVIEWER COMMENTS

Reviewer #2 (Remarks to the Author):

The authors have provided detailed responses to my comments. I appreciate the addition of additional experiments. Considering the simplifications of the physical model when treating curvature and potential misunderstandings regarding the membrane-protein curvature (my comments 2.11 to 2.15 and referee #1 comments 1.5 to 1.7) and the various measurement uncertainties, I would abstain from strong quantitative statements, and rather keep the message qualitative as state-dependent protein sorting according to membrane geometries.

Response:

Our reasoning for making quantitative statements based on our simplified physical model is listed below:

1. We clearly state in the revised manuscript the assumptions and potential limitations of our model.

2. We clearly present the uncertainties (standard errors) associated with each of our measurements.

3. All experimental measurements require certain models and assumptions to be interpreted. If one measures the size of Piezo1 with EM or AFM, the data will very likely have higher *precision* compared to our light microscopy measurements. However, it doesn't mean the data will have a higher *accuracy* to represent the size of Piezo1 as a biological channel. This is for the simple reason that one needs to assume that the Piezo1 being measured in froze liposomes/detergents via EM/AFM represents the Piezo1 channel in living cell membranes. In comparison, our 'less precise' measurement is done on a more accurate target.